



# Optically thin clouds in the trades

Theresa Mieslinger[1], Bjorn Stevens[2], Tobias Kölling[2], Manfred Brath[1], Martin Wirth[3], and Stefan A. Buehler[1]

[1]Universität Hamburg, Faculty of Mathematics, Informatics and Natural Sciences, Department of Earth Sciences, Meteorological Institute, Hamburg, Germany
[2]Max Planck Institute for Meteorology, Hamburg, Germany
[3]Institut für Physik der Atmosphäre, Deutsches Zentrum für Luft- und Raumfahrt e.V., Wessling, Germany

**Correspondence:** Theresa Mieslinger (theresa.mieslinger@mpimet.mpg.de)

**Abstract.** We develop a new method to describe the total cloud cover including optically thin clouds in trade wind cumulus cloud fields. Climate models as well as Large Eddy Simulations commonly underestimate the cloud cover, while estimates from observations largely disagree on the cloud cover in the trades. Currently, trade wind clouds contribute significantly to the uncertainty in climate sensitivity estimates derived from model perturbation studies. To simulate clouds well and especially
how they change in a future climate we have to know how cloudy it is.

In this study we develop a method to quantify the cloud cover from a clear-sky perspective. Using well-known radiative transfer relations we retrieve the clear-sky contribution in high-resolution satellite observations of trade cumulus cloud fields during EUREC[4]A. Knowing the clear-sky part, we can investigate the remaining cloud-related contributions consisting of areas detected by common cloud masking algorithms and those undetected areas related to optically thin clouds. We find that the
cloud-mask cloud cover underestimates the total cloud cover by a factor of 2. Lidar measurements on board the HALO aircraft support our findings by showing a high abundance of optically thin clouds during EUREC[4]A. Mixing the undetected optically thin clouds into the clear-sky signal can cause an underestimation of the cloud radiative effect of up to -32%. We further discuss possible artificial correlations in aersol-cloud cover interaction studies that might arise from undetected optically thin clouds. Our analysis suggests that the known underestimation of trade wind cloud cover and simultaneous overestiamtion of cloud
brightness in models is even higher than assumed so far.

## 1  Introduction

Earth's trade wind regions combine a dry atmosphere and a high abundance of shallow clouds – whose tops are often not much higher than the long-wave emission height – to efficiently cool the planet. How much clouds in the trades cool the climate is quantified by their cloud radiative effect, which in a first approximation depends on the cloud cover and the average cloud
reflectance. Changes in the cloud radiative effect with warming pace cloud feedbacks, which in the trades have been shown to contribute significantly to uncertainties in estimates of the global climate sensitivity (Bony and Dufresne, 2005; Vial et al., 2016), part of the well known difficulty climate models have in representing clouds and cloud changes with fidelity.





Especially in low-cloud regions such as the trades, climate models underestimate the cloud cover while overestimating it's average reflectance, a problem often called the "too few, too bright" low-cloud problem (Nam et al., 2012; Klein et al., 2013).

Large eddy simulation studies also show an underestimation of trade wind cumulus cloud cover and a limited representation of small clouds (Nuijens et al., 2015), while the scaling behaviour of trade cumulus clouds suggests a high abundance and significant contribution of small clouds to the total cloud cover (Plank, 1969; Wielicki and Welch, 1986; Cahalan and Joseph, 1989; Benner and Curry, 1998; Zhao and Di Girolamo, 2007; Mieslinger et al., 2019). Studies on the "twilight" zone even suggest that clouds may extend further into the cloud-free area than assumed so far (Koren et al., 2008). To simulate the change

in clouds with future temperature or aerosol perturbations, we first need to know how cloudy it is.

Estimating the cloud cover is a well-known issue in the sense that it decisively depends on the instrument used and the purpose of respective datasets. All-sky observations by trained humans might have been the first systematic cloud-cover measurements. Such measurements are synonymous with efforts to predict the weather and led to the first International Cloud Atlas as early as 1896. However, such observations are subject to unknown or hard to quantify uncertainties due to the training of

the observer and further biases originating from overlapping cloud layers and undetected upper clouds, or the higher frequency of fair weather synoptic reports (Warren et al., 1985). Passive remote sensing opened the way to more objective quantification of cloud cover from ground, from aircraft since the beginning of the 20th century, and also from space starting in the 1970s. Active remote sensing added additional approaches to investigate clouds from ground, aircraft, and from space. Those various instruments dedicated to observe clouds have in common the dependence of a best estimate of cloud cover on (a) the data

resolution in space and / or time, (b) suitable thresholds defined in the physical quantity closest to the instrument raw data, (c) the wavelength used and the resulting sensitivity of the measurement to clouds. Even for collocated measurements with very high spatial (tens of meters) and temporal resolution, Fig. 5 in Stevens et al. (2019) and more recently Konow et al. (in prep.) nicely show that the range of cloud cover estimates from active and passive remote sensing can differ by a factor of 2.

In this study we present a different view on clouds by quantifying the clear-sky area. The clear-sky signal is well understood

in radiative transfer relations and can be simulated with well-posed approximations. The main advantage of estimating cloudiness as the complement to clear-sky is that we overcome the problem of diverse and instrument-specific hard-coded thresholds in cloud masking algorithms. We apply the clear-sky approach to high-resolution satellite imagery from the Advanced Spaceborne Thermal Emission and Reflection Radiometer (ASTER) recorded during the field campaign EUREC$^4$A in Jan-Feb 2020. EUREC$^4$A was dedicated to the investigation of trade wind cumulus clouds and their interaction with the large-scale envi-

ronment (Bony et al., 2017; Stevens et al., 2021). The high resolution of the ASTER data provides the possibility to include clouds of sizes at the deca- to hectometer scale and, equally important, increases the probability to observe clear-sky pixels free of any cloud structures. With the clear-sky approach we can detect enhanced reflectances from anomalously humidified aerosols and optically thin cloud areas that are undetected by traditional cloud-masking algorithms. We show the contribution of optically thin cloud areas to the total cloud area and use Lidar measurements on board the HALO research aircraft to support

our findings.

The remainder of this article is organized as follows. Section 2 describes the high-resolution ASTER satellite dataset, the WALES Lidar cloud product, and surface wind speed data based on ERA5 reanalysis. In Section 3 we show the clear-sky





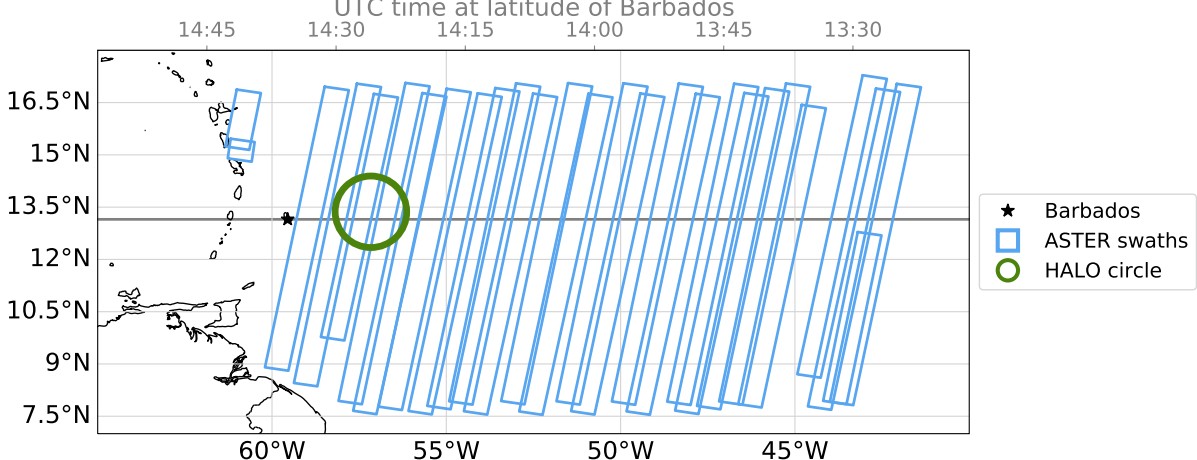

**Figure 1.** ASTER dataset during EUREC[4]A with 419 images (60km x 60km) recorded on 17 days between 11 January and 19 February 2020. WALES lidar measurements are available from HALO's research flights predominantly on the circular path shown in green from 13 flight days between January 22 and February 15 2020.

model setup, and how we identify optically thin clouds in ASTER observations. Results on the contribution of optically thin clouds to the total cloud cover during EUREC[4]A are shown in Section 4, followed by a discussion of implications of our results in Section 5.

## 2 Observations

Within this study we exploit the potential of the high spatial resolution passive remote sensing instrument ASTER (Advanced Spaceborne Thermal Emission and Reflection Radiometer; Yamaguchi et al. (1998)) that recorded images of cloud fields east of Barbados in support of the EUREC[4]A campaign. We extend the information on the typical cloud fields observed during EUREC[4]A with airborne high spectral resolution lidar measurements to support our analysis of clouds from an active sensor with a high sensitivity to small and optically thin clouds.

### 2.1 The ASTER dataset for EUREC[4]A

ASTER is mounted aboard Terra, a polar-orbiting satellite in a Sun-synchronous orbit with an equator crossing time of 10:30 local solar time. Terra crosses the latitude of Barbados and the HALO flight circle area roughly at 14:25 UTC, while the tracks further east at about 43°W are observed by ASTER an hour earlier. Fig. 1 shows the measurements taken in the area east of Barbados from 7 °N to 18 °N and from 41 °W to 62 °W between January 11 and February 19 2020. The data from the observed swaths are segmented in the form of $60 \times 60 \ km^2$ images, each corresponding to 9 s of observation time.





ASTER's visible and near-infrared (VNIR) radiometer pointing nadir has three bands in the range of 0.53 - 0.86 $\mu$m. The radiometrically calibrated and geometrically co-registered Level 1B data provide top of atmosphere monodirectional radiances

at 15 m pixel resolution at the sub satellite point. We use the band 3 radiance centered at 0.807 $\mu$m in the present study to define the total cloud cover. We further draw comparisons to the ASTER cloud mask which is based on several bands in the VNIR. The cloud mask works with thresholding tests and is representative for traditional passive remote sensing cloud masking schemes. In detail, we distinguish between *confidently clear*, *probably clear*, *probably cloudy*, and *confidently cloudy* pixels following the method described in Werner et al. (2016) for the VNIR bands. Within this study we combine the flags *probably*

*cloudy* and *confidently cloudy* if we refer to cloudy regions according to the ASTER cloud mask. We omit thresholding tests including the broken short-wave infrared detector as well as ASTER's thermal band 14 (11.65 $\mu$m, 90 m pixel resolution). The latter would detect cirrus contaminated areas at the expense of a lower resolution.

In our analysis we work with reflectances instead of radiances with the aim to reduce the influence of varying solar zenith angles $\theta_0$ within the overpasses and slightly varying extraterrestrial solar irradiance $E_0$. The reflectance $R$ is calculated from

the radiance $L$ as

$$R = \frac{\pi L}{cos(\theta_0) E_0} \tag{1}$$

## 2.2  WALES airborne lidar measurements

The WALES lidar instrument (Water Vapor Lidar Experiment in Space demonstrator; Wirth et al. (2009)) is part of the remote

sensing package on board the HALO research aircraft during EUREC[4]A (Stevens et al., 2019). The high spectral resolution lidar measurements from the auxiliary channels of the instrument at 532 nm are well suited to investigate the small and optically thin clouds due to the high instrument sensitivity to small particles ranging from aerosols to cloud droplets. The advantage of WALES compared to space borne active instruments such as the Cloud-Aerosol Lidar with Orthogonal Polarization (CALIOP) simply lies in the closer distance and thus a higher sensitivity to low clouds and the much higher horizontal sampling due to the

lower aircraft speed (0.2 km/s versus 7 km/s). The resulting horizontal spatial resolution of the WALES cloud product is about 40 m during EUREC[4]A, which is slightly larger but commensurate with that of ASTER. CALIOP has been shown to struggle detecting small clouds with cloud tops below 1 km (Leahy et al., 2012), while we find 29 % of clouds detected by WALES during EUREC[4]A to have cloud tops below 1 km.

Within the present study we use the cloud mask and cloud optical depth product described in Konow et al. (in prep.). In the

dataset, a cloud is defined where the backscatter ratio exceeds 10. This threshold is lower compared to the studies by Gutleben et al. (2019) and Jacob et al. (2020) where the value was chosen to make the detection limit comparable to CALIOP. The lower value used in the present study nicely separates the highest possible signals originating from marine aerosol and any cloud related signal that might include anomalously humidified aerosols and the smallest cloud droplets. WALES uses the High Spectral Resolution Lidar technique (HSRL; Esselborn et al. (2008)) to distinguish molecular from particle backscatter

at 532 nm, which allows for the direct measurement of the (two way) atmospheric transmission. The latter is proportional to





the range $(r)$ and atmospheric density corrected lidar signal $R_M(r)$. To a first approximation the optical thickness is given by

$$\tau = -\frac{1}{2} \cdot \ln\left(\frac{R_M(r)}{R_M(0)}\right). \tag{2}$$

The complete algorithm adds several corrections and is described in detail in Esselborn et al. (2008).

## 2.3 Surface wind speed estimates

For the methodology described in Sec. 3 we need surface wind speed estimates at 10 m height for a given ASTER pixel. The fifth generation European Centre for Medium-Range Weather Forecasts reanalysis (ERA5) provides hourly wind speed estimates on a global grid at 10 m height (2D surface product) which would fit our needs, but showed a significant underestimation compared to collocated dropsonde measurements during EUREC⁴A (JOANNE dropsonde dataset: George et al. (under review)). The underestimation is in agreement with a study by Belmonte Rivas and Stoffelen (2019) which find a low bias

in ERA5 surface winds in the trades. Nevertheless, wind speed estimates from the ERA5 profile product (hourly, 0.25° grid; Hersbach et al. (2020)) agree remarkably well with dropsonde measurements.

Thus, we use ERA5 wind speeds at the lowest pressure level 1000 hPa which corresponds to about 135 m above sea level on average based on the dropsonde dataset. We derive a correction that translates from 1000hPa to 10m based on a comparison of ERA5 wind speed at 1000 hPa and the 10 m wind speed from dropsonde measurements (Pearson correlation coefficient 0.88).

A least squares fit provides us with the coefficients to estimate the 10 m wind speed by

$$ws = 0.92 \cdot ws_{ERA5,1000hPa} + 0.40. \tag{3}$$

This wind speed is an average value representative for a 0.25° grid cell. We therefore use measurements at the Barbados Cloud Observatory (BCO) to estimate the variance in wind speed within 0.25° compared to the 15 m ASTER grid. The BCO is located at the easternmost point of the island of Barbados and has been shown to take measurements representative of an

undisturbed marine trade wind boundary layer (Stevens et al., 2016). We use the standard surface wind speed measurements from a Vaisala WXT-520 to derive an estimate of the surface wind variance within 0.25° (27.12 km at 13 °N) which translates to about 80 minutes sampling period. We add a Gaussian perturbation according to the estimated wind variance of $1.63\,\mathrm{m^2s^{-2}}$ to the average wind speed within our further analysis. The campaign average wind speed corresponding to the ASTER image locations is 9.02 ms⁻¹.

## 130 3 Methodology

The ASTER cloud mask provides us with a good perception of the certainly clear and certainly cloudy areas, while we are less confident in between. We approach the intermediate range from the clear-sky by simulating the expected probability distributions of clear-sky reflectances for a given ASTER image. Knowing the theoretical clear-sky contribution to an all-sky ASTER image we can then investigate the cloud-related contributions that are undetected by the cloud mask and which we

attribute to optically thin clouds.





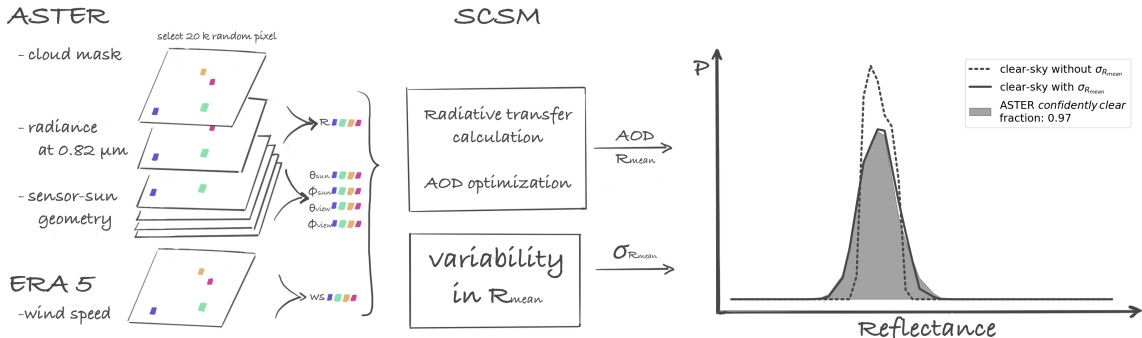

**Figure 2.** Sketch illustrating the clear-sky retrieval workflow. ASTER and ERA5 input data is used to run radiative transfer simulations with integrated AOD optimization. A Gaussian perturbation is added to the output average pixel reflectance $R_{mean}$ to account for ocean surface variability and measurement noise. The figure on the right shows the processing steps that lead to the simulated clear-sky reflectance distribution for a single ASTER image observed 2020-01-24 14:02:02 UTC.

We start with a brief overview on the clear-sky retrieval setup and the necessary input information on surface wind speed and aerosol optical depth, before we show our approach for transferring the clear-sky information to the ASTER observations and defining areas of optically thin clouds.

### 3.1 A simplified clear-sky model (SCSM)

The clear-sky radiance over ocean in the visible range depends on a narrow set of parameters and can be estimated by simplified 1D radiative transfer calculations. In appendix A we describe the full set of equations and approximations made in calculating the clear-sky signal with our simplified clear-sky model (SCSM). We generally assume a single-layer atmosphere with constant air density and calculate the extinction of solar radiance from the top of atmosphere to the ground and back to the sensor in space. How the light is reflected at the surface into the view direction of the sensor is characterized by the bi-directional

reflection function which depends on the surface wind speed and the generated ocean wave slope distribution. Here, we use the wind speed estimates described in chapter 2.3 as input to the Cox and Munk parameterization to derive an average reflectance for a given surface condition.

We further need to know the aerosol optical depth (AOD) to estimate the extinction of direct and diffuse light on it's path through the atmospheric column. Although the aerosol load does not vary much within a 60 x 60 km$^2$ ASTER image, the

availability of aerosol information from measurements even for an image-average AOD is very limited. Therefore, we estimate an effective AOD in an optimization approach by including information from the ASTER dataset. We assume that the pixels labeled *confidently clear* in the ASTER cloud mask are a good first guess for clear-sky and shall serve as a reference for finding a suitable effective AOD such that the simulated clear-sky values are in close agreement with the selected ASTER pixel values.



In Fig. 2 we illustrate the clear-sky retrieval workflow. In detail, we randomly select 20000 pixel from those defined *confidently clear* by the ASTER cloud mask (see Sect. 2.1) for a given ASTER image. Simulating 20000 samples ensures a proper representation of the clear-sky distribution at a manageable computational cost. For those input pixel locations we run the clear-sky model with the corresponding sensor-sun geometries, surface wind speed estimates, and a first guess on the AOD. We further optimize this image AOD value iteratively by minimizing the summed squared difference between simulated and observed reflectances. Here, we make use of scipy's implementation of the limited-memory Broyden–Fletcher–Goldfarb–Shanno algorithm (LM-BFGS) with bounds (scipy version 1.5.2). From all evaluated ASTER images we find a campaign average effective AOD of 0.076 ($\pm$ 0.051).

From comparing simulated clear-sky reflectance distributions to the observed ones for fully clear-sky ASTER observations we find two things. First, the distributions agree very well in terms of their expected value. Second, the simulated distributions are more narrow compared to the observed ones as the Cox and Munk parametetrization returns average pixel reflectances $R_{mean}$. We therefore introduce a variability in brightness in a post processing step. We calculate a kernel density estimate with normal kernels characterized by a standard deviation $\sigma_{R_{mean}}$ that is placed on each of the simulated reflectance values (Rosenblatt, 1956; Parzen, 1962). We derive a suitable value for $\sigma_{R_{mean}}$ from comparing simulated clear-sky reflectance distributions and corresponding ASTER images that have at minimum 97 % *confidently clear* pixels in the ASTER cloud mask. From 22 cases we calculate the average $\sigma_{R_{mean}} = 0.0026$ from a least-squares optimization using again the LM-BFGS algorithm. We use a constant value for $\sigma_{R_{mean}}$ for the whole dataset due to the lack of several clear-sky observations for various sensor-sun geometries. However, the ASTER dataset is confined to a narrow set of sensor-sun geometries and outside of possible sun glint observations such that we assume that a constant value is sufficient for our application.

## 3.2 Identifying optically thin clouds in all-sky observations

The output from our SCSM model provides us with a distribution of clear-sky reflectances $p(R|F_{\text{CLEAR}}, B)$, which is the probability distribution of reflectance values $R$ given that they originate from clear-sky area with the flag $F = F_{\text{CLEAR}}$ and additional background conditions $B$. The background conditions include the sensor-sun geometry, wind speed, and AOD and are covered by the SCSM by handling each image individually. In the following we evaluate the probabilities on an image basis and therefore omit the implicit condition on $B$ in the notation. Further, we use standard notation whereby "|" means "given that" for conditional probabilities and "," means "and" and symbolizes combined (or joint) probabilities. For example, the SCSM output is a conditional probability as the SCSM framework does not include any information on the general clear-sky fraction within one image.

In the following, we split the observed reflectance distribution of an ASTER image into the categories or flag values $F \in \{F_{\text{CLEAR}}, F_{\text{OTC}}, F_{\text{CLOUD}}\}$. The ascending order of the flag values indicates the associated expected increase in reflectance. The darkest observed pixels originate form clear-sky ocean observations. Small cloud fragments and humidified aerosols slightly enhance the reflectance, though they are often undetected by cloud masking scheme. We characterize them as optically thin clouds OTC. The flag CLOUD refers to the cloudy pixels detected by the ASTER cloud masking scheme (see Sec. 2.1). We know the CLOUD part of an distribution $p(R, F_{\text{CLOUD}})$ from the observation and we can infer the CLEAR contribution from the


SCSM output. The all-sky reflectance distribution $p(R)$ is build up by the arithmetic sum of combined probability distributions of $R$ and the flag values $F$, that is:

$$p(R) = \sum_{F_n} p(R, F_n)$$ (4)
$$= p(R, F_{\text{CLEAR}}) + p(R, F_{\text{OTC}}) + p(R, F_{\text{CLOUD}})$$

Each combined probability can be represented by the product of the corresponding conditional probability and the probability of the flag value, i.e. for clear-sky

$$p(R, F_{\text{CLEAR}}) = p(R|F_{\text{CLEAR}}) \cdot p(F_{\text{CLEAR}}).$$ (5)

The probability of clear-sky $p(F_{\text{CLEAR}})$ is the true clear-sky fraction in an observed image and challenging to estimate. Note that the true clear-sky fraction is independent of the ASTER cloud mask. If we would know the clear-sky fraction $p(F_{\text{CLEAR}})$, equations Eq. 5 and Eq. 4 together fully describe the observed reflectance distribution $p(R)$. In the following we describe our approach for estimating the unknown clear-sky fraction.

The first constraint is given by the fact that any probability must be within the range $[0,1]$, thus we can formulate for our
case:

$$p(F_{\text{CLEAR}}|R'') + p(F_{\text{CLOUD}}|R'') \leq 1 \qquad \forall R'' \in R$$ (6)

We can approach the estimation of the clear-sky fraction $p(F_{\text{CLEAR}})$ from a conservative side by deriving the maximum possible $p(F_{\text{CLEAR}})$ such that Eq.6 still holds. Thinking visually, we scale the simulated clear-sky distribution up until it touches the all-sky distribution $p(R)$. At the reflectance $R = R'$ (of unknown value) where the PDFs touch, we are certain that the non-cloudy
classified reflectances are actually due to clear-sky:

$$\exists \, R' \text{ such that } p(F_{\text{CLEAR}}|R') = 1 - p(F_{\text{CLOUD}}|R')$$ (7)

We can solve Eq. 7 and Eq. 6 for $p(F_{\text{CLEAR}})$ (for details see appendix B). While being mathematically concise, the described method faces a problem. It relies on the exact count of measurements in only a single reflectance bin $R'$ and thus is especially susceptible to measurement and model uncertainties. We tackle this problem by extending and relaxing the condition stated
in Eq. 7. We modify this first condition from a single value to an extended range of reflectance values. As Eq. 7 would be overdetermined for more than one reflectance value in the presence of measurement and model uncertainties, we demand that the equation approximates the value $1 - p(F_{\text{CLOUD}}|R')$ for reflectivity values measured and known to be caused by clear-sky.

In particular, we do this by a weighted linear regression, minimizing the term:

$$\int |[p(F_{\text{CLEAR}}|R) - (1 - p(F_{\text{CLOUD}}|R))] \cdot w|^2 \, dR$$ (8)





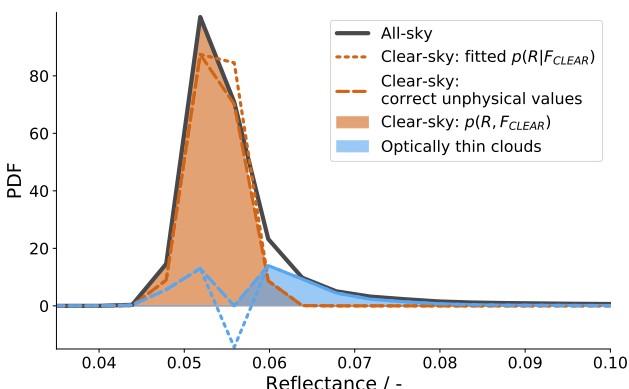

**Figure 3.** Visualization of the approach for estimating the clear-sky fraction $p(F_{\text{CLEAR}})$ by optimization. The orange dotted and dashed lines show the processing steps leading to the filled orange clear-sky PDF. The blue lines are the respective residuals related to optically thin clouds and resulting from the all-sky (grey) minus the CLEAR (orange) and minus the CLOUD PDF (dark blue; not visible).

with $p(F_{\text{CLEAR}})$ as the only free variable. The regression weight $w = p(R)p(R|F_{\text{CLEAR}})$ is chosen to only consider measured reflectances $p(R)$ that overlap with the range of simulated clear-sky reflectances $p(R|F_{\text{CLEAR}})$. The product of both guarantees a close agreement around the peaks of measured and simulated PDF.

The resulting estimate of $p(F_{\text{CLEAR}})$ is more robust in the presence of small measurement or model errors, but a direct consequence of this approximate matching is that Eq. 6 does not necessarily hold for all $R''$ anymore. As illustrated in Fig. 3
using dotted and dashed lines, we correct this by clipping the resulting probabilities to the allowed range. As this clipping effectively modifies the simulated reflectance distribution and thus is potentially dangerous, we need to ensure that this method indeed only compensates for small measurement uncertainties (i.e. in the order of a single digital sensor count). We can do this by comparing the expected value of the clear sky reflectance $p(R|F_{\text{CLEAR}})$ before and after clipping. On average, this difference is $0.15\%$ and even in the worst (maximum) case, the clipping causes a shift of $0.0018$ in reflectance units, which is well below
one digital sensor count of about $0.004$ reflectance units. Based on this analysis, we use the more stable regression and clipping method in stead of a direct application of Eq. 7.

Further, the SCSM does not include cloud shadows on the ocean surface which introduce a signal at very low reflectances in the observed distribution. Conceptually we add the low reflectance values originating from such shadowed areas to the clear-sky reflectance distribution $p(R, F_{\text{CLEAR}})$.

In Fig. 4 we show combined probability distributions per flag for an ASTER observation on the 31$^{\text{st}}$ of January east of Barbados. The inset figure shows the reflectance image that we translate into the distribution using the method described above.





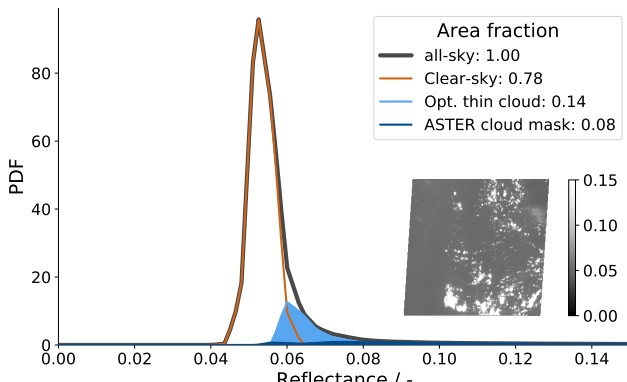

**Figure 4.** Reflectance distribution corresponding to the ASTER observation shown in the inset figure recorded on 31 January 2020, 14:08:05 UTC south-east of the HALO circle area at 11.37 °N, 53.86 °W. The clear-sky contribution is retrieved with the method (1) described in section 3.2 and displayed by the orange curve, while pixel reflectances identified cloudy from the ASTER cloud masking algorithm are shown in dark blue. We attribute light blue contribution to the distribution to optically thin clouds.

## 3.3 Robustness of optically thin cloud estimation

Our target variables are the fraction and expected reflectance of optically thin clouds. The retrieval of clear-sky and subsequent
optically thin clouds in ASTER images depends on visible clear-sky areas which limits the evaluation of the full ASTER EUREC$^4$A dataset to images with less than 85 % detected cloud cover in the cloud masking algorithm (395 images).

Within the retrieval we have two main free parameters which can introduce uncertainty in our target values, the surface wind speed estimate and the assumed variability $\sigma_{R_{mean}}$ of simulated average pixel reflectances $R_{mean}$. We first have a look at the added variability. From a comparison of 22 manually checked clear-sky reflectance distributions ($> 97$ % *confidently*
*clear* pixels) to the simulated distributions we derived an average variance of 0.0026 ($\pm 0.0007$). We apply the methodology described in this section for the average value, as well as for a 20 % lower (0.0020) and 20 % higher value (0.0031). Similarly, we add an artificial bias of $\pm 20$ % to the surface wind speed estimates and investigate the change in our target values. The average wind speed in our dataset is 9.02 ms$^{-1}$ ($\pm 2.38$ ms$^{-1}$). The resulting deviations in our target values, the fraction $p(\text{OTC})$ and expected reflectance $E(R|\text{OTC})$ of optically thin clouds, that result from a bias in $\sigma_{R_{mean}}$ and / or the surface wind speed
are stated in Tab. 1 and Tab. 2.

The fraction of optically thin clouds $p(\text{OTC})$ changes only slightly with a change in wind speed showing an overestimation for a negative wind speed bias meaning that a small part of the clear-sky distribution is wrongly attributed to optically thin clouds. For a positive wind speed bias the opposite is the case. The low uncertainties (3.1 % and -2.5 %) are a result of the retrieval setup including the optimization of AOD which can partly compensate a bias in wind speed. Changing the variability
of simulated average pixel reflectances $\sigma_{R_{mean}}$ can narrow (negative bias in $\sigma_{R_{mean}}$) and broaden (positive bias in $\sigma_{R_{mean}}$) the clear-sky distribution and thus lead to strong over- or underestimation of $p(\text{OTC})$ as high as $\sim 10$ %. Combining the highest



**Table 1.** Deviations of the fraction of optically thin clouds $\Delta p$(OTC) for the two main free parameters to the clear-sky retrieval, the surface wind speed and the variability $\sigma_{R_{mean}}$. The two numbers in each cell state the absolute / relative difference to the reference case with no wind speed bias and $\sigma_{R_{mean}} = 0.0026$ respectively.

| | | wind speed bias | | |
| --- | --- | --- | --- | --- |
| | | -1.8 ms$^{-1}$ | 0 ms$^{-1}$ | 1.8 ms$^{-1}$ |
| | 0.0020 | 0.026 / 14.2% | 0.018 / 10.1% | 0.012 / 6.3% |
| $\sigma_{R_{mean}}$ | 0.0026 | 0.006 / 3.1% | 0 / 0% | -0.005 / -2.5% |
| | 0.0031 | -0.012 / -6.4% | -0.019 / -10.0% | -0.022 / -11.9% |

**Table 2.** Deviations of the expected reflectance of optically thin clouds $\Delta E(R|\text{OTC})$ for the two main free parameters to the clear-sky retrieval, the surface wind speed and the variability $\sigma_{R_{mean}}$. The two numbers in each cell state the absolute / relative difference to the reference case with no wind speed bias and $\sigma_{R_{mean}} = 0.0026$ respectively.

| | | wind speed bias | | |
| --- | --- | --- | --- | --- |
| | | -1.8 ms$^{-1}$ | 0 ms$^{-1}$ | 1.8 ms$^{-1}$ |
| | 0.0020 | -0.0043 / -4.4% | -0.0031 / -3.1% | -0.0017 / -1.8% |
| $\sigma_{R_{mean}}$ | 0.0026 | -0.0011 / -1.1% | 0 / 0% | 0.0014 / 1.4% |
| | 0.0031 | 0.0024 / 2.5% | 0.0038 / 3.9% | 0.0049 / 5.0% |

retrieval uncertainties from the two free parameters, the wind speed and the variability $\sigma_{R_{mean}}$, we can get a deviation in the estimated fraction of optically thin clouds of $\pm 0.026$ ($\pm 14.2\,\%$).

The expected reflectance of optically thin clouds $E(R|\text{OTC})$ shows a smaller sensitivity to changes in the wind conditions
and $\sigma_{R_{mean}}$ compared to the fraction of optically thin clouds discussed above. An underestimation in wind speed leads to a marginal underestimation in the expected reflectance as lower clear-sky reflectances are wrongly attributed to optically thin clouds. In the case of an overestimation in wind speed, the clear-sky reflectance distribution extends to higher reflectance values which are missing in the estimated $E(R|\text{OTC})$ and thus leads to a high bias in $E(R|\text{OTC})$. A more narrow (negative bias in $\sigma_{R_{mean}}$) or broader (positive bias in $\sigma_{R_{mean}}$) clear-sky distribution can decrease or increase the expected reflectance of
optically thin clouds up to $\pm 4\,\%$. However, the combined deviation due to possible biases in wind speed and $\sigma_{R_{mean}}$ are still within the range of $\pm 0.0049$ ($\pm 5.0\,\%$).

## 4 Results

We investigate 395 ASTER images for the signal from optically thin clouds (OTC) that are undetected by the ASTER cloud mask but can be identified with the method described in Sect. 3. We first visualize pixels in an image that we attribute to
the total cloud cover including OTC pixels and those detected in the ASTER cloud mask. We then define a close match of





OTC reflectances in ASTER images and the signal of OTC detectable in WALES lidar data. WALES measurements provide an independent view of the results of the cloud cover by OTC from a different instrument technology and complement our analysis based on ASTER images. Finally, we show the significant contribution of optically thin clouds to the total cloud cover.

### 4.1 Visualizing optically thin clouds in an ASTER image

To visualize the OTC area in an image we can define a threshold in reflectance similar to common cloud masking algorithms. We construct a total cloud cover mask that includes pixels with a probability of that pixel reflectance to be cloudy $p(F_{\text{TOTAL\_CLOUD}}|R = R_{\text{pixel}}) \geq 0.9$ with $F_{\text{TOTAL\_CLOUD}} = F_{\text{OTC}} \vee F_{\text{CLOUD}}$. In the particular ASTER image shown partially in Fig. 5 all reflectance values greater than 0.049 satisfy that condition. The cloud mask derived with the cloud masking algorithm by including several ASTER bands is shown in blue in panel a) while the total cloud cover mask is shown by the contours in

red in panel b). The background reflectance image in panel b) is adjusted in its reflectance range with the aim to enhance the range reflectances related to OTC.

    The figure visualizes how OTC is often classified in pixels surrounding detected clouds. Detraining clouds and anomalously humidified aerosols likely cause enhanced reflectances close to thicker clouds. Possible scattering of light at the sides of thicker clouds might additionally enhance the brightness of their surrounding areas. Such surrounding halos of optically thin clouds

lead to (threshold dependent) smoother cloud edges, an interesting result in the context of cloud boundaries and related fractal dimensions. Also, cloud structures tend to be more connected in the total cloud cover mask leading to larger cloud objects with smooth reflectance transitions to the clear-sky ocean background. While there are numerous studies on cloud shapes we rather focus on a statistical estimate of area coverage and the contribution of OTC to the total cloud cover in the remainder of this work.

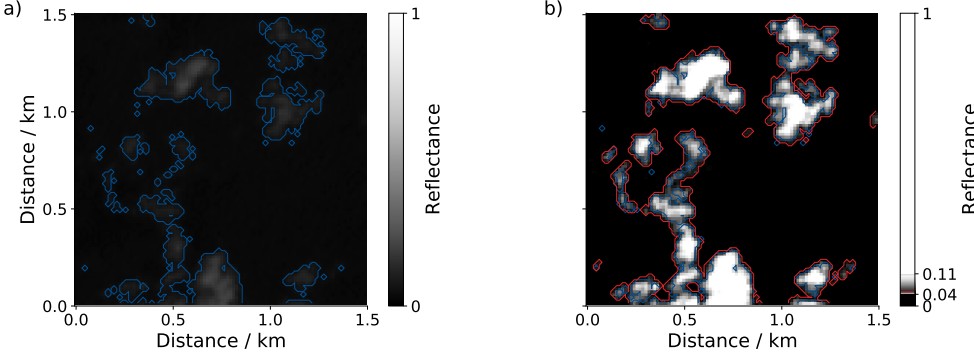

**Figure 5.** Visualization of the area corresponding to optically thin clouds. Shown are reflectances at 0.807 $\mu$m for a 1.5 x 1.5 km$^2$ selection of an ASTER image recorded on 5 February 2020, 14:25:15 UTC. (a) shows the full physical range of reflectance values ranging from 0 to 1 with overlayed blue contours outlining the ASTER cloud mask. (b) is similar to (a) but with the color scale limited to the 10$^{\text{th}}$ and 90$^{\text{th}}$ percentile of reflectances attributed to total cloud cover including optically thin clouds. The red contours correspond to $p(F_{\text{TOTAL\_CLOUD}}|R = R_{\text{pixel}}) \geq 0.9$.





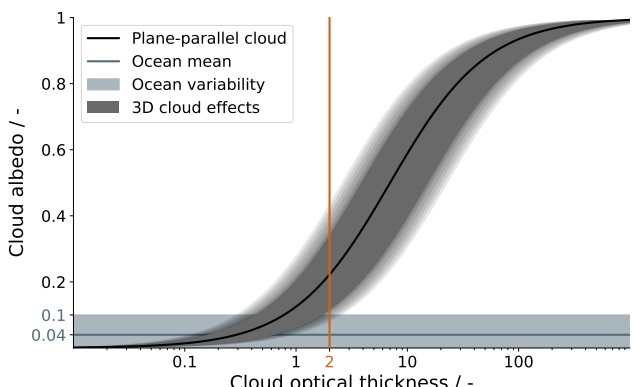

**Figure 6.** Plane-parallel relationship between cloud albedo and cloud optical thickness following Lacis and Hansen (1974). The ocean reflectance is estimated from the ASTER observations during EUREC$^4$A, while the uncertainty due to 3D radiative effects is a rough estimate from the literature.

## 4.2 The OTC equivalence in Lidar data

In Fig. 5 optically thin clouds are barely visible in the reflectance field in panel a) suggesting that those clouds have a very low cloud optical thickness. Due to non-linearities in the physical and radiative properties of small cumulus clouds and the large influence of 3D radiative effects, plane-parallel retrieval of microphysical properties do not work reliably and we cannot derive cloud optical thickness from ASTER measurements directly (Davies, 1978; Loeb et al., 1997; Várnai and Marshak, 2003; Marshak et al., 2006; Stevens et al., 2019; Kölling, 2020). However, we use the theoretical relationships that plane-parallel retrievals are based on to estimate an effective cloud optical thickness that could be detected by ASTER against the ocean surface background following the two-stream approximation by Lacis and Hansen (1974):

$$A = \frac{\sqrt{3}(1-g)\tau}{2+\sqrt{3}(1-g)\tau} \approx \frac{\tau}{\tau+7.7} \tag{9}$$

with the cloud albedo $A$, cloud optical thickness $\tau$ and the asymmetry parameter $g = 0.85$. In Fig. 6 we show the relationship stated in Equ. 9 of a plane-parallel cloud (black line) and add uncertainties from cloud 3D effects and the background ocean signal.

The average ocean reflectance during EUREC$^4$A was 0.04 including single cases as high as 0.07. Due to additional variability in the ocean wave reflection we expect that clouds with an albedo below 0.1 and corresponding cloud optical thickness below 1 to dissolve in the ocean signal. For clouds with cloud optical thickness larger than 1, 3D effects such as brightening and shadowing as well as photon loss through the cloud sides become relevant and can easily cause a factor of 2 error in the reflectance (Marshak et al., 2006; Stevens et al., 2019). We therefore assume that due to natural variability in the background ocean signal and the cloud signal, clouds with optical thickness below 2 do not stand out clearly from the ocean and the ASTER cloud mask presumably is insensitive to such optically thin clouds.




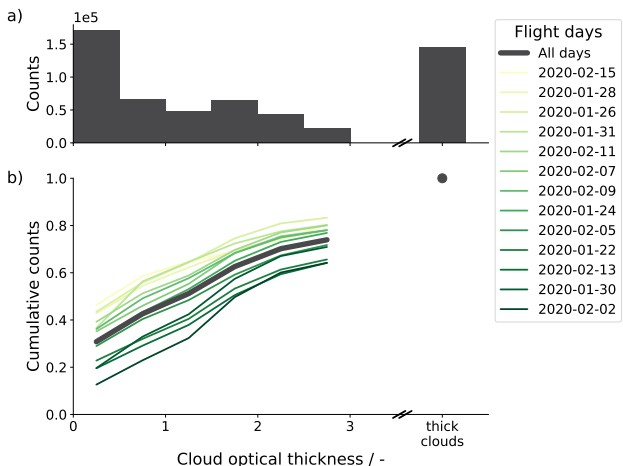

**Figure 7.** Cloud optical thickness distribution from WALES lidar measurements for all days with local research flights during EUREC[4]A resulting in 92 hours of data. Panel a) shows the frequency distribution of all days, while panel b) additionally shows the cumulative distributions for individual days. The days are sorted by their increasing average cloud optical thickness that we associate with optically thin clouds (yellow to dark green). The split x-axis visualizes the limited information on thick clouds that are optically opaque to the lidar.

Clouds with an optical thickness below 2 are thin enough for a lidar beam to penetrate through the cloud and provide a reliable estimate of the cloud optical thickness. We can therefore make use of WALES lidar measurements for supporting information on the abundance of optically thin clouds.

Fig. 7 shows the distribution of cloud optical thickness measurements from WALES for days with local research flights. The peak at low cloud optical thickness values corresponds to optically thin clouds that the lidar beam manages to penetrate. A cloud with optical thickness of about 2.5 reduces the lidar signal below the cloud to more than one hundredth and the method to derive the optical thickness still works. At night the range of retrieved optical thickness increases to about 3.5 due to a better signal to noise ratio above clouds without scattered sun light. In thicker clouds the signal vanishes in the system noise. We aggregate all measurements from optically opaque and thick clouds in one bin as we have no information on the actual cloud optical thickness.

In WALES measurements we associate optically thin clouds to have an optical thickness below 2. The campaign average cloud optical thickness of OTC is 0.75, the median is 0.52. Optically thin clouds have on average a cloud top height at 1.5 km altitude (median 1.2 km). We further use the WALES measurements to derive a fractional cloud cover in time for optically thin clouds and compare the results to the optically thin cloud cover from ASTER in the following section.

### 4.3 The contribution of OTC to the total cloud cover

From analysing 395 ASTER images during EUREC[4]A we find an average total cloud cover of 42 %, combined of 24 % from detected clouds and 19 % from optically thin clouds (see Tab. 3). Based on the clear-sky retrieval uncertainties derived in





**Table 3.** Cloud cover estimates during EUREC[4]A from 395 ASTER satellite observations ($60 \times 60\,\text{km}^2$) at 15 m resolution on 17 days and from WALES lidar measurements recoded within 13 research flights (days) at about 40 m resolution in January and February 2020.

|  | Optically thin cloud cover / % | Total cloud cover / % |
|---|---|---|
| ASTER (mean) | 19 | 42 |
| ASTER (median) | 19 | 36 |
| WALES (mean) | 21 | 34 |

Sec. 3.3 we estimate the uncertainty in ASTER optically thin cloud cover to be within the range of $\pm\,2.6\,\%$. In Table 3 we state the respective numbers derived from WALES measurements. We explicitly note that a direct comparison is not reasonable as the two instruments and approaches show optically thin cloud areas from two different perspectives. However, what we can say is that WALES lidar measurements indicate a high fractional coverage by optically thin clouds, similar to what we find from ASTER images.

In Sec. 4.1 we mentioned the possible influence of scattering at cloud edges which can illuminate areas surrounding thicker clouds. Such 3D effects would influence our results based on ASTER data and lead to an overestimation of OTC related cloud cover. As WALES is less affected by the 3D scattering at cloud edges but shows a higher fraction of optically thin clouds relative to ASTER, the ASTER analysis does not seem to be unduly influenced by 3D radiative effects.

Our results based on ASTER and WALES measurements are lower compared to an analysis of optically thin marine clouds from CALIOP measurements by Leahy et al. (2012). From two years of nighttime measurements the authors attribute 45 % of total cloud cover to optically thin clouds between 60 °S and 60 °N, while in the trades the fraction of optically thin clouds is as high as 84 %. From WALES measurements we derived an OTC fraction of 63 % for cloudy profiles with cloud optical thickness < 2. If we include clouds with cloud optical thickness up to about 3 as it is done in the study by Leahy et al. (2012), the OTC fraction in WALES data increases to 74 %. Estimates based on CALIOP data are likely to overestimate the OTC fraction due to the lower sensor resolution of 90 m footprints every 335 m. The authors in Leahy et al. (2012) derive a possible overestimation of OTC fraction of up to 25 % in the trades due to partially cloudy CALIOP footprints, which supports our findings in the current study of a lower, but still significant contribution of optically thin clouds to the total cloud cover.

We further notice that the area covered by optically thin clouds increases with detected cloud cover for low total cloud cover as shown in Fig. 8 and similarly stated in Leahy et al. (2012). The positive correlation up to 0.4 total cloud cover might be due to a combination of two features. First, optically thin cloud areas are often found surrounding detected clouds (see also Fig. 5). This idea is supported in a study by Koren et al. (2007), which find enhanced reflectances in solar irradiance measurements before and after an identified cloud originating from humidified aerosols and/or unresolved cloud fragments.

The second ingredient to the proposed positive correlation is the cloud field structure. Trade wind cumulus cloud fields at low cloud cover typically correspond to *sugar* or *gravel* type structures as described by Stevens et al. (2020), consisting of many small clouds with enough space in between that can be partly filled with undetected optically thin clouds. More clouds





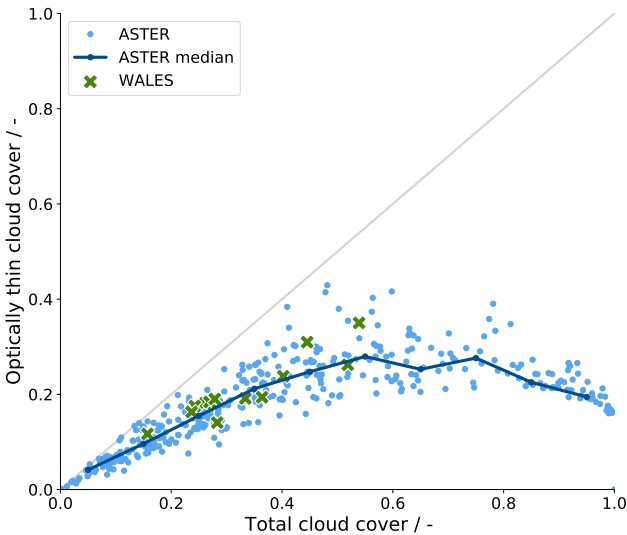

**Figure 8.** Change in optically thin cloud cover with total cloud cover. The blue markers correspond to values derived from 395 ASTER images ($60 \times 60 \, \text{m}^2$) with the dark blue line following along the median values. The green markers correspond to daily-averaged cloud cover estimates from WALES lidar measurements. The grey diagonal line shows the maximum possible contribution of optically thin clouds to the total cloud cover.

and more cloud boundary therefore leads to more optically thin cloud area up to a point where this relationship saturates at about 0.4 total cloud cover. The saturation might be due to larger clouds or cloud structures being surrounded by pronounced clear-sky regions. A recent study by Schulz et al. (2021) identifies the so-called *flower* and *fish* cloud patterns of having characteristic clear-sky areas between clouds. By constraint, the positive correlation turns negative above 0.7 total cloud cover as the clear-sky, OTC, and detected cloud cover always add up to 1 and high cloud-mask cloud cover situations leave little space for optically thin clouds.

We conclude that optically thin clouds cover large parts of the trades leading to a higher total cloud cover than assumed so far from passive satellite observations.

## 4.4 The cloud reflectance - cloud cover relationship in ASTER observations

Current climate models typically have a narrow range of cloud optical thickness that might affect model perturbation experiments due to the non-linearity of cloud optical thickness and it's albedo. Especially in low-cloud regions such as the trades, climate models underestimate the cloud cover while overestimating it's average reflectance, a problem often called the "too few, too bright" low-cloud problem (Nam et al., 2012; Klein et al., 2013). While observations show a positive correlation of cloud cover and cloud reflectance, models show a reverse sign (Konsta et al., 2016).

We investigate the cloud cover - cloud reflectance relationship in Fig. 9 and Fig. 10. Fig. 9 panel a) shows in blue curves the change in all-sky reflectance distribution with increasing cloud cover as defined by the ASTER cloud mask, while the red



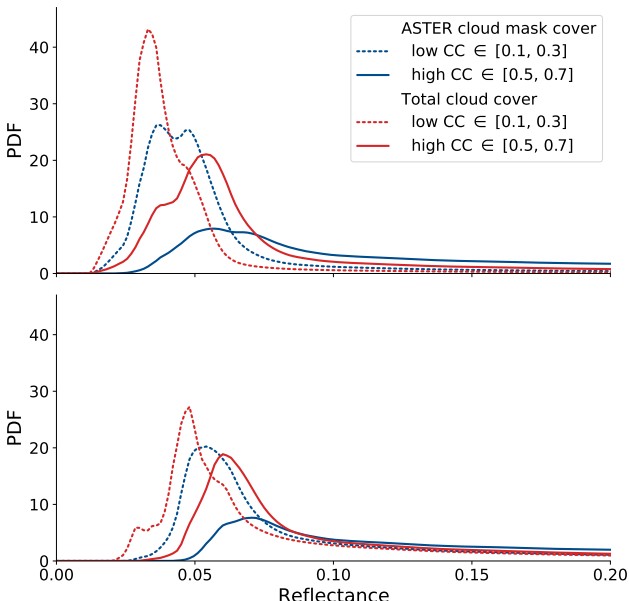

**Figure 9.** Combined probability density functions (PDF)s of a) all-sky reflectances from ASTER p(R|CC), binned according to the total (red) and cloud mask (blue) cloud cover (CC). We define two representative cloud cover ranges, low CC (0.1 to 0.3) and high CC (0.5 to 0.7). Panel b) shows the conditional probability of total cloud reflectances p(R|F$_{TOTAL}$, CC), given that they are within the range of low or high CC. Compared to a), the distributions in panel b) do not include the clear-sky contributions at low reflectances.

lines show similarly the change with increasing total cloud cover. We show two representative cloud cover ranges, a low range from 0.1 to 0.3 and a high range from 0.5 to 0.7. With increasing cloud cover, the reflectance distributions shift to higher values meaning that the overall image is brighter. As expected, the total cloud cover reflectance distributions peak at lower reflectances compared to their cloud-mask counterparts meaning that the total cloud cover area is less bright on average.

Panel b) shows an interesting new facet to the difference in total and cloud-mask cloudy areas. The distributions show how the total cloud reflectance relative to the total cloud area in the image depends on cloud cover. The comparison of low and high cloud cover cases reveals that clouds are brighter with increasing cloud cover, which is in agreement with our perception of larger, deeper, and brighter clouds being present in high cloud cover situations. The change in cloud brightness with cloud cover is less pronounced if the total cloud cover is considered (including optically thin clouds) compared to the cloud-mask only case.

We further investigate the expected cloud reflectances in relation to derived cloud cover values for all 395 ASTER images in Fig. 10. Both, cloud mask and total cloud cover, exhibit positive correlations with respective cloud reflectance values in agreement with findings in Konsta et al. (2016). We here derive a campaign average cloud reflectance from total cloud cover of 0.15, with contributions from cloud-mask clouds (avg: 0.21) and optically thin clouds (avg: 0.10), which agrees quite well with an average trade wind cumulus cloud reflectance of 0.15 derived from a combination of POLDER (Polarization and Directionality of the Earth's Reflectances) and CALIOP measurements in the study by Konsta et al. (2016). Based on the



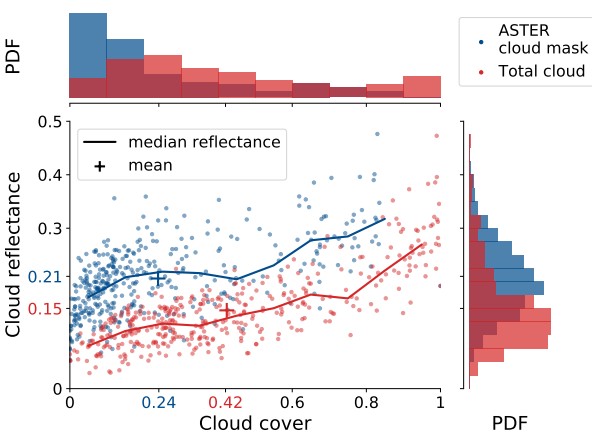

**Figure 10.** Expected cloud reflectance corresponding to the ASTER cloud mask (blue) and the derived total cloud cover (red) from 395 ASTER images. The median cloud reflectances are given by the lines and the dataset averages are visualized by the "+" marker and the respectively colored tick labels. The frequency distributions of cloud cover and cloud reflectance are shown in the panels on the top and right respectively.

clear-sky retrieval uncertainty stated in Sec. 3.3, the uncertainty in expected reflectance of optically thin clouds is as low as
380   0.005 and does not influence our results and conclusions drawn here.

    The positive correlation in Fig. 10 for total cloud cover agrees well with the corresponding Fig. 6a in Konsta et al. (2016). As mentioned before, climate models show a reverse sign of this correlation together with a general underestimation of cloud cover and simultaneous overestimation of cloud reflectance. Next to the model intrinsic mechanisms leading to too few, but too bright clouds, biases might be partially due to tuning the model based on traditional cloud masks that overestimate the cloud
385   reflectance especially in the frequent low cloud cover situations.

## 5   Discussion

Most passive satellite imagers operate at resolutions in the order of hectometer to kilometer range and derive cloud products at 1 km scale or coarser. Undetected optically thin clouds, as well as small clouds detected at the ASTER 15 m scale, are unresolved and lead to partially cloudy pixel measurements. Several studies in the past have investigated the resolution effect
390   in trade cumulus cloud cover estimated from passive satellite imagers. Zhao and Di Girolamo (2006) find a three- to fivefold overestimation of cloud cover in MODIS and MISR images respectively compared to ASTER observations during the RICO campaign. For the same dataset, a study by Dey et al. (2008) suggests a fourfold overestimation of cloud cover if the ASTER cloud mask is degraded from 15 m to 1 km while cloud detection thresholds are kept constant. However, degrading the resolution can also lead to an underestimation of cloud cover estimates in cloud masking schemes if the resulting pixel radiances fall





below fixed radiance thresholds. In an early study by Wielicki and Parker (1992) the authors estimate that roughly one third
of the cloud cover detected in 30 m Landsat images showing cumulus clouds would not be detected by certain cloud masking
schemes, which is in line with our study results.

An underestimation of cloud cover due to undetected optically thin clouds and an overestimation due to an reduced spatial
resolution have compensating tendencies. However, one effect that does not cancel out in typical passive satellite cloud products
is the influence of optically thin clouds in partially cloudy pixels that are classified to be clear. Pure clear-sky observations are
crucial for aerosol retrievals, as well as cloud radiative effect (CRE) estimates. With decreasing sensor resolution the probability
for clear-sky observations decreases as well. We therefore investigate implications that undetected optically thin clouds can
have on CRE estimates, as well as our inferences on cloud-aerosol interactions in the trades, despite their low cloud albedo.

## 5.1 Implication for CRE estimates

In temperature perturbation studies, cloud feedback defines how clouds adjust to a perturbation in surface temperature and
whether this change amplifies or dampens the initial temperature perturbation. As such, it is tied to the cloud radiative effect
(CRE), the difference in all-sky and clear-sky radiative flux at the top of the atmosphere, in the initial as well as in the perturbed
climate.

$$CRE = F_{\text{ALL}} - F_{\text{CLEAR}} \tag{10}$$

In the trades, climate models show a less negative CRE in response to warming, indicative of a positive cloud feedback (Zelinka
et al., 2020). Observational constraints based on satellite data at coarse resolution might be insensitive to sub-pixel scale clouds
and consequently lack a robust clear-sky signal. From our analysis we can estimate an upper bound on the error in CRE that
might arise from a clear-sky signal that is contaminated by undetected optically thin clouds.

If we assume that the pixel reflectances corresponding to optically thin clouds from the present analysis are fully mixed
into the clear-sky signal, we would overestimate the clear-sky reflectance and consequently underestimate the CRE. We de-
rive a relative bias $\Delta CRE$ per image from the differences in all-sky $L_{\text{ALL}}$, clear-sky $L_{\text{CLEAR}}$, and "contaminated" clear-sky
$L_{\text{CLEAR+OTC}}$ expected radiance values:

$$\Delta CRE = \frac{CRE_{\text{CLEAR+OTC}} - CRE_{\text{CLEAR}}}{CRE_{\text{CLEAR}}} \tag{11}$$

$$= \frac{L_{\text{ALL}} - L_{\text{CLEAR+OTC}}}{L_{\text{ALL}} - L_{\text{CLEAR}}} - 1 \tag{12}$$

Note that we use here the simulated clear-sky $L_{\text{CLEAR}}$ radiances as those do not contain the low radiances from cloud shadows
on the ocean surface which would cause a slight underestimation of the clear-sky radiance.

In principle, a mono-directional radiance $L$ can be converted to a radiative flux $F$ as it is done by Clouds and the Earth's
Radiant Energy System (CERES) radiative flux products by the following equation (Loeb et al., 2003; Su et al., 2015):

$$F = \frac{\pi L(\theta_s, \theta_v, \Phi)}{f(\theta_s, \theta_v, \Phi)} \tag{13}$$





with the sun $\theta_s$ and sensor view $\theta_v$ zenith angles, the azimuthal difference $\Phi$ and the anisotropic factor $f$. The anisotropic factor is challenging to estimate and no suitable values are available for ASTER observations. However, if we assume isotropic scattering of cumulus cloud fields ($f = 1$) we can translate the CRE bias into an effective radiative flux at 0.807 $\mu$m.

The mean CRE bias from the ASTER dataset is as high as -32 % which roughly translates to about -6 $\mathrm{Wm}^{-2}$. The order of magnitude of the potential CRE bias from optically thin clouds is comparable to the magnitude of the aerosol direct effect that

has been estimated to be about 5 $\mathrm{Wm}^{-2}$ for the winter trades in Loeb and Manalo-Smith (2005) highlighting the importance of an improved representation of optically thin clouds in future studies.

## 5.2 Optically thin clouds in the aerosol-cloud interaction context

First, we would like to point out the difference between optically thin clouds and aerosols. The marine boundary layer is a humid layer with the constant presence of humidified sea-salt and ammonium sulfate aerosols. The mixing within the boundary

layer will bring the aerosols almost always into an environment above 80 % relative humidity such that sea-salt and ammonium sulfate deliquesce, while the humidity is almost everywhere above 60 % making it impossible for the aerosols to effloresce (humidity as shown by the JOANNE dropsonde dataset, George et al. (under review)). Thus, humidified aerosols are omnipresent and part of the clear-sky signal. As both, ASTER and WALES data suggest a total cloud cover well below 100 % (insensitive to the exact cloud threshold in WALES) we are confident that the described signal of optically thin clouds can only be due to

anomalously humidified aerosols and cloud droplets.

Aerosol-cloud interaction studies are a topic in itself and we will not go into great detail, but rather want to show where optically thin clouds might need to be considered in these studies. One largely debated issue is the positive correlation of AOD and cloud cover as an indirect aerosol effect. The underlying principle is that hydrophilic aerosols can serve as cloud condensation nuclei and increase the cloud droplet number concentration. More aerosols might therefore reduce the precipitation formation

rate and increases the cloud liquid water content and cloud lifetime (Albrecht, 1989). Whether this so-called cloud lifetime effect actually leads to increased cloud cover is largely debated (Loeb and Manalo-Smith, 2005; Kaufman et al., 2005; Stevens and Feingold, 2009; Gryspeerdt et al., 2016).

While modeling studies suggest negligible or equally small enhancing or decreasing influences of aerosols on the cloud cover (Xue and Feingold, 2006; Quaas et al., 2008; Seifert et al., 2015), observational studies mostly rely on coarse satellite

observations and show deficiencies in the accuracy in aerosol and cloud retrievals as discussed in Quaas et al. (2020). The positive correlation in optically thin cloud cover and detected clouds in the current study suggests that part of the proposed sensitivity of cloud cover to AOD might reflect a high bias in clear-sky estimates that is interpreted as high AOD. In agreement with our perception, an observational study by Gryspeerdt et al. (2016) estimates meteorological covariations to account for 80 % of the often proposed AOD-cloud cover relationship with the additional note on shallow cumulus regions having a very

weak relationship.

Independent of the cloud-lifetime effect, a positive perturbation in aerosols increases the cloud droplet number concentration and thus the cloud brightness, which is commonly referred to as the Twomey effect (Twomey, 1959; Quaas et al., 2020). Increasing the brightness also increases the probability of undetected and optically thin clouds identified in the current study





to cross the detection threshold of common cloud masking schemes. We therefore speculate that the Twomey effect indirectly

leads to positive AOD-cloud cover relationships found in previous studies. It might be interesting to investigate the AOD-cloud cover relationship based on a more comprehensive definition of total cloud cover including optically thin clouds.

## 6  Conclusions

Climate models as well as Large Eddy Simulations commonly underestimate the cloud cover, while estimates from observations largely disagree on the cloud cover in the trades. We use a new method to estimate the total cloud cover from the clear-sky

perspective by simulating the clear-sky contribution to an observed all-sky reflectance distribution with a simplified radiative transfer model. The present study shows the high abundance of optically thin clouds in the trade wind region that are undetected by common cloud-masking schemes.

We analyzed 395 ASTER satellite images recorded in support of the EUREC[4]A field campaign in January and February 2020 and find that about half of the total cloud cover is due to undetected optically thin clouds. A comparison to independent

WALES lidar measurements supports our findings.

We find that pixels attributed to optically thin clouds are often found surrounding brighter cloud objects that can be detected in cloud-masking schemes. Accounting for optically thin clouds significantly ($29\pm2\%$) reduces the average cloud reflectance as optically thin clouds are systematically less reflective than clouds detected in cloud masking schemes. Our analysis suggests that the known underestimation of trade wind cloud cover and simultaneous overestimation of cloud brightness in models is

even higher than assumed so far.

We identify two implications from our study. First, if mixed into the clear-sky signal, the enhanced radiance from optically thin cloud areas leads to a high bias in clear-sky estimates over ocean and hence a low bias up to -32 % in the estimated cloud radiative effect of trade wind cumulus cloud fields.

And second, the positive correlation in optically thin cloud cover and detected clouds for low cloud cover suggests that part of

the sensitivity of cloud cover to AOD found in aerosol-cloud interaction studies might reflect a high bias in clear-sky estimates that is interpreted as high AOD. In addition, increasing cloud brightness with higher AOD likely increases the probability of undetected and optically thin clouds identified in the current study to cross the detection threshold of common cloud masking schemes. These effects could contribute to an unrealistically strong relationship between satellite retrieved values of AOD and cloud cover, and would suggest that not accounting for optically thin clouds could overstate the strength of aerosol cloud

interactions.

*Code and data availability.*  In addition to the publicly available ASTER L1B data from NASA we provide processed data for the ASTER images recorded during EUREC[4]A and displayed in Fig. 1. NetCDF files containing physical quantities from bands in the VNIR and thermal range, latitude and longitude information, a cloud mask, and cloud top height estimates are available on the AERIS data server (https://observations.ipsl.fr/aeris/eurec4a-data/SATELLITES/TERRA/ASTER/). ASTER image tiles were calculated and are stored on AERIS

(https://observations.ipsl.fr/aeris/eurec4a/Leaflet/index.html) providing a user-friendly browsing experience with the possibility to zoom



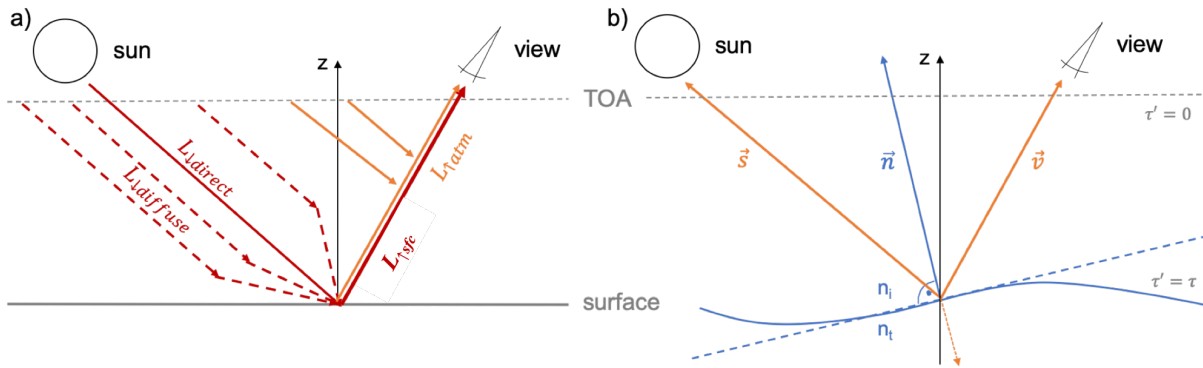

**Figure A1.** Sketches of the simple clear-sky model. a) illustrated the main radiance components, while b) shows the geometry setup based on the vectors $s$ pointing into the sun, $v$ pointing to the sensor, and the wave facet normal $n$.

in on the rich structures of beautiful trade cumulus cloud fields. The cloud information from WALES is published on AERIS https://doi.org/10.25326/216) and further described in Konow et al. (in prep.).

Code for processing the original ASTER L1B data is available in the Python package *typhon* version 0.8.0, subpackage *cloudmask* (https://github.com/atmtools/typhon). The basic code for the clear-sky radiative transfer simulations is available at https://doi.org/10.5281/
zenodo.4842675. The main data resulting from the applied methodology and forming the basis for all interpretations is available at https://doi.org/10.5281/zenodo.4844482.

## Appendix A: Components and equations to the simple clear-sky model (SCSM)

Knowing the extraterrestrial irradiance $E_0$ emitted by the sun and entering the atmosphere, the radiative transfer equation describes the radiance at any location (x, y, z) and for any direction defined by a zenith angle $\theta$ and an azimuthal angle $\phi$. In
a clear-sky atmosphere with small solar and viewing zenith angles we can use 1D plane-parallel radiative transfer to estimate the radiance observable at the top of atmosphere (TOA).

The clear-sky radiance $L$ reaching a sensor in space is a combination of three main components that we illustrate in Fig. A1 a): (1) the direct sun ray reflected at the ocean surface $L_{\downarrow \text{direct}}$ and (2) the hemispheric diffuse radiance reflected at the surface towards the sensor $L_{\downarrow \text{diffuse}}$. Together they are combined in the component $L_{\uparrow sfc}$ of light that touched the surface. On the way
from the surface to the sensor $L_{\uparrow sfc}$ experiences attenuation following Lambert-Beer and depending on the atmospheric optical thickness $\tau$ and the cosine of the sensor or view zenith angle $v_z$. In addition, there is component (3), the diffuse light from single-scattering events happening within the atmosphere $L_{\uparrow atm}$.

$$L = L_{\uparrow \text{sfc}} + L_{\uparrow \text{atm}} \tag{A1}$$

$$= \exp\left(\frac{-\tau}{v_z}\right)\left[L_{\downarrow \text{direct}} + L_{\downarrow \text{diffuse}}\right] + L_{\uparrow atm} \tag{A2}$$





In the following, we describe the derivation of $L$ based on the vector $\boldsymbol{s}$ pointing from an observed location on the ground to the sun, and the view vector $\boldsymbol{v}$ pointing to the sensor (see Fig. A1 b)).

$$\boldsymbol{s} = \begin{pmatrix} s_x \\ s_y \\ s_z \end{pmatrix}, \boldsymbol{v} = \begin{pmatrix} v_x \\ v_y \\ v_z \end{pmatrix} \tag{A3}$$

$\boldsymbol{s}$ and $\boldsymbol{v}$ are unit vectors meaning that they satisfy the condition:

$$\mid \boldsymbol{s} \mid = \mid \boldsymbol{v} \mid = 1. \tag{A4}$$

Working with vectors instead of the traditional approach with angles simplifies several of the following calculations next to a significant enhancement in computational speed. For example, the previously mentioned view zenith angle $v_z$ is simply the third component of the view vector $\boldsymbol{v}$.

**A1   Direct radiance and the bi-directional reflection function (BRDF)**

$L_{\downarrow\text{direct}}$ is defined by the sensor-sun geometry with the cosine of the sun zenith angle $s_z$ and the corresponding aerosol extinction
along the path from the top of atmosphere (TOA) to the surface where the reflection is characterized by the bi-directional reflection function (BRDF) $\rho$.

$$L_{\downarrow\text{direct}} = E_0 \exp\left(\frac{-\tau}{s_z}\right) \rho(\boldsymbol{s}, \boldsymbol{v}, ws, n_i, n_t) \tag{A5}$$

How a sun ray is reflected at the ocean surface mostly depends on the surface wind speed $ws$ and the generated wave slopes. The earliest and still widely used surface slope parametrization goes back to photographic measurements by Cox and Munk
in 1954. Their parametrization is embedded in a 1D Guassian surface slope distribution $p$, combined with Fresnel reflection coefficients for unpolarized light $r$ and a prefactor handling the sensor-sun geometry with the sun $\boldsymbol{s}$ and view $\boldsymbol{v}$ vectors. For the general equation for $\rho$ we follow Stamnes et al. (2017):

$$\rho(\boldsymbol{s}, \boldsymbol{v}, ws, n_i, n_t) = \frac{1}{4 v_z s_z (n_z)^4} \cdot p(\boldsymbol{s}, \boldsymbol{v}, ws) \cdot r(\boldsymbol{s}, \boldsymbol{v}, n_i, n_t) \tag{A6}$$

In the first factor, $n_z$ is the third component of the wave facet normal $\boldsymbol{n}$ with

$$\boldsymbol{n} = \begin{pmatrix} n_x \\ n_y \\ n_z \end{pmatrix} = \frac{\boldsymbol{s} + \boldsymbol{v}}{\mid \boldsymbol{s} + \boldsymbol{v} \mid} \tag{A7}$$

The second factor in Eq. A6 gives the probability of a specular reflection $p$ and the third the intensity of the reflected light $r$. In detail, we assume a 1D Guassian surface slope probability distribution $p$ with

$$p(\boldsymbol{s}, \boldsymbol{v}, ws) = \frac{1}{\pi\sigma(ws)^2} \exp\left(-\frac{1 - n_z^2}{n_z^2 \cdot \sigma(ws)^2}\right) \tag{A8}$$





and the variance $\sigma^2$ of the surface slope distribution. The Cox and Munk parametrization provides an empirical estimate for $\sigma^2$

depending on the 10 m surface wind speed $ws$ (Cox and Munk, 1954):

$$\sigma(ws)^2 = 0.003 + 0.00512 \cdot ws. \tag{A9}$$

The intensity of the reflected light $r$ is given by the unpolarized Fresnel reflection coefficient:

$$r(\boldsymbol{s}, \boldsymbol{v}, n_i, n_t) = \frac{1}{2}\left[\left(\frac{\mu_i - n_r \mu_t}{\mu_i + n_r \mu_t}\right)^2 + \left(\frac{\mu_t - n_r \mu_i}{\mu_t + n_r \mu_i}\right)^2\right] \tag{A10}$$

with $n_r = \frac{n_t}{n_i}$, the ratio of the refractive index of the transmitted medium $n_t = 1.333$ (ocean) and the refractive index of the

incoming medium $n_i = 1$ (atmosphere). Further, $\mu_i$ is the cosine of the incidence angle and is given by the dot product of the

sun and wave facet normal vector:

$$\mu_i = \boldsymbol{s} \cdot \boldsymbol{n} \tag{A11}$$

$\mu_t$ is the cosine of the transmission angle, which follows directly from Snell's law by transformation:

$$\mu_t = \sqrt{1 - \frac{1 - \mu_i^2}{n_r^2}} \tag{A12}$$

**A2  Diffuse downward radiance and hemispheric BRDF**

The hemispheric diffuse radiance $L_{\downarrow \text{diffuse}}$ includes sun rays that are scattered within the atmosphere on their way to the ground

and get reflected at the pixel of interest into the direction of the sensor view. Thus, we integrate the integration vector $\boldsymbol{x}$ over

the hemisphere $\Omega$:

$$L_{\downarrow \text{diffuse}} = \int_\Omega \rho(\boldsymbol{x}, \boldsymbol{v}, ws) \cdot L_{\text{in}}(\tau, \boldsymbol{x}) d\boldsymbol{x} \tag{A13}$$

Assuming that the incoming diffuse downward radiance $L_{\text{in}}(\tau, \boldsymbol{x})$ is isotropic, we can pull $L_{in}$ out of the integral and derive

a hemispheric BRDF by integrating equation A6 over $\Omega$. Here, we make use of the Gauss-Legendre quadrature to approximate

the integral based on only a few nodes in the $\mu$ space while keeping a high accuracy.

The diffuse downward irradiance on the other hand is difficult to approximate. Thus, we sample from a pre-calculated

look-up table of diffuse downward irradiance for a range of sun zenith angles and aerosol optical depths. The look-up table

was calculated with the full radiative transfer model libRadtran for a sensor at the surface pointing up nadir and observing

at ASTER's band 3 central wavelength 807 nm (Mayer and Kylling, 2005; Emde et al., 2016). The input file defines a U.S.

Standard Atmosphere with default molecular absorption calculated with the representative wavelengths parameterization REP-

TRAN (medium) where the absorption is based on the HITRAN 2004 catalog. The aerosols species is set to be *maritime*

*tropical* as defined by the OPAC package and finally, the radiative transfer equation is solved with DISORT. We further use the

bivariate spline approximation provided within the Python package *scipy (version 1.5.2)* to interpolate over the output look-up

table.





### A3 Diffuse upward radiance from single-scattering events

The atmospheric diffuse scattering $L_{\uparrow atm}$ describes sun rays that are reflected within the atmosphere into the view direction of the sensor. We only consider single scattering events as the aerosol optical depth over tropical ocean is mostly below or in the

order of 0.1 and the probability of further scattering events is unlikely. The extinction within an atmospheric column is generally given by the integral over the extinction coefficients $\sigma_{ext,i}$ in single atmospheric layers depending on their density (temperature) and particles. We simplify the problem by integrating over $\tau$ instead of the atmospheric path lengths with $dl = \frac{dz}{cos(\theta)}$ of a respective zenith angle $\theta$. Correspondingly, we can write the integral over all single (aerosol) scattering events along an atmospheric path $l$ from the surface to TOA

$$L_{\uparrow atm} = E_0 \int\limits_{sfc}^{TOA} \exp\left(-\frac{1}{s_z}\int\limits_{z_{scat}}^{TOA} \sigma_{ext}(z)dz\right) \tag{A14}$$

$$\cdot \exp\left(-\frac{1}{v_z}\int\limits_{z_{scat}}^{TOA} \sigma_{ext}(z)dz\right) \tag{A15}$$

$$\cdot \sigma_{scat}\Theta_{HG}\, dz_{scat} \tag{A16}$$

where the extinction is accounted for in the exponential functions with the scattering event happening at the height $z_{scat}$. The product of the scattering coefficient $\sigma_{scat}$ and the scattering phase function $\Theta_{HG}$ describes the scattering efficiency.

In our atmospheric column of constant density, $\sigma_{scat}$ is independent of height and the integral $\int_{sfc}^{TOA} \sigma_{scat}\, l\, dl$ simplifies to $\int_0^\tau d\tau'$ with $\tau$ being the optical depth of the atmospheric column. In more detail, we can rewrite the relation and include the single scattering albedo $\omega_0$

$$\sigma_{scat} \cdot dl = \omega_0 \sigma_{ext} \cdot \frac{dz}{\mu_v} = \frac{\omega_0}{\mu_v}d\tau \tag{A17}$$

and further include those in Eq. A14:

$$L_{\uparrow atm} = \Theta_{HG}\frac{\omega_o}{\mu_v}\int\limits_0^\tau \exp\left(\frac{-\tau'}{\mu_o}\right)\exp\left(\frac{-\tau'}{\mu_v}\right)d\tau' \tag{A18}$$

The Henyey Greenstein phase function $\Theta_{HG}$ is an approximation for the scattering phase function and only depends on the assymetry parameter $g$, that is the mean cosine of the scattering angle calculated by integrating over the scattering phase function (Henyey and Greenstein, 1941):

$$\Theta_{HG} = \frac{1}{4\pi}\frac{1-g^2}{1+g^2-2g\left(\mu_{scat}\right)^{3/2}} \tag{A19}$$

For $\omega_0$ and $g$ we use constant values taken from the libRadtran calculations with the input setup described in Sect. A2.





## Appendix B: Derivation of the clear-sky fraction

Based on equations Eq. 6 and Eq. 7 we could directly solve for the clear-sky fraction $p(F_{\text{CLEAR}})$.

We start with the clear-sky model output and apply Bayes' theorem:

$$p(F_{\text{CLEAR}}|R) = \frac{p(R|F_{\text{CLEAR}})}{p(R)} \cdot p(F_{\text{CLEAR}}) \tag{B1}$$

We can add this information to Eq. 7

$$1 - p(F_{\text{CLOUD}}|R') = \frac{p(R=R'|F_{\text{CLEAR}})}{p(R=R')} \cdot p(F_{\text{CLEAR}}) \tag{B2}$$

and solve for $p(F_{\text{CLEAR}})$

$$p(F_{\text{CLEAR}}) = \frac{p(R=R')}{p(R=R'|F_{\text{CLEAR}})}(1 - p(F_{\text{CLOUD}}|R')) \tag{B3}$$

We further add the information from Eq. B1 and Eq. B3 to our constraint stated in Eq. 6:

$1 \geq p(F_{\text{CLEAR}}|R) + p(F_{\text{CLOUD}}|R) \tag{B4}$

$$1 \geq \frac{p(R|F_{\text{CLEAR}})}{p(R)} \cdot \frac{p(R=R')}{p(R=R'|F_{\text{CLEAR}})}(1 - p(F_{\text{CLOUD}}|R')) \tag{B5}$$

$$+ p(F_{\text{CLOUD}}|R) \tag{B6}$$

Rearranging the equation we get

$$\frac{p(R=R'')}{p(R=R''|F_{\text{CLEAR}})}(1 - p(F_{\text{CLOUD}}|R'')) \tag{B7}$$

$$\geq \frac{p(R=R')}{p(R=R'|F_{\text{CLEAR}})}(1 - p(F_{\text{CLOUD}}|R')) \qquad \forall R'' \in R \tag{B8}$$

and consequently we can find R' by searching for the minimum:

$$R' = argmin_{R''}\left(\frac{p(R=R'')}{p(R=R''|F_{\text{CLEAR}})} - (1 - p(F_{\text{CLOUD}}|R''))\right) \tag{B9}$$

Knowing the R' we could in principle derive the clear-sky fraction p($F_{\text{CLEAR}}$) from Equ. B3. However, Equ.B9 becomes unstable where p($F_{\text{CLOUD}}$|R") is close to 1 which corresponds to cloudy parts while we are interested in the clear part of the 605   distribution. We therefore apply the modified method described in Sec. 3.2 in the current study.

*Author contributions.* BS and TM conceptualized the study. TK, MB, and TM worked out the methodology. MW derived the lidar cloud optical thickness data. SB and BS supervised the project. TM conducted the analysis, and prepared the manuscript with contributions from all co-authors.

*Competing interests.* The authors declare that they have no conflict of interest.





*Acknowledgements.* This study was supported by the International Max Planck Research School on Earth System Modelling (IMPRS-ESM), Hamburg, and the Universität Hamburg. It also contributes to the Cluster of Excellence "CLICCS—Climate, Climatic Change, and Society" funded by DFG (EXC 2037, Project Number 390683824), and to the Center for Earth System Research and Sustainability (CEN) of Universität Hamburg.

The ASTER L1B data product was retrieved from the online Data Pool, courtesy of the NASA Land Processes Distributed Active Archive

Center (LP DAAC), USGS/Earth Resources Observation and Science (EROS) Center, Sioux Falls, South Dakota (Last accessed 2021-04-21 from https://doi.org/10.5067/ASTER/AST_L1B.003). We would like to thank the ASTER Science Team for scheduling the ASTER data acquisition in support of the EUREC[4]A field campaign. We acknowledge ECMWF and the Copernicus Climate Change Service for providing access to the ERA5 data set through the Climate Data Storage API (last accessed 2021-04-21). Theresa Mieslinger would like to thank Bernhard Mayer for his feedback on a suitable libRadtran setup, as well as Jean-Louis Dufresne for helpful discussions on the cloud

cover - cloud reflectance relationship in climate models.



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
