# Peer review of "Optically thin clouds in the trades"

_Atmospheric Chemistry and Physics, 2021_

## Author Comment (AC1)

**Answers to reviewers**

**General**

We would like to thank the reviewers for the time and effort devoted to critically assessing our manuscript. The request raised in Review #2 for a direct comparison of the flags used within our study and the ones used by the ASTER cloud mask following Werner et al., 2016 (see **R2\_C3**) as well as the question **R1\_C6** of Figure 4, ended up being especially helpful. When implementing the additional flag dimension in our work flow we detected a "bug" in the sense that we were including swath edge pixels. As a consequence the fractional coverage of cloud - OTC and cloud-free areas were wrongly normalised. The revised manuscript has therefore different numbers and some statements change in their strength. However, and most importantly, none of the overall conclusions change, that it, that optically thin clouds are important :)

Another (smaller) change results from excluding potentially cirrus-contaminated images from the analysis. Please see the answer to comment 1 by reviewer #1 (**R1\_C1**) below.

Further, the question by reviewer #3 (**R3\_C5**) made us reconsider our conclusion drawn from Figure 6 that, based on plane-parallel retrieval theory, clouds with optical thickness below 1 will have a high probability of being undistinguishable from the ocean surface albedo. Also, the appearance of clouds in satellite images can vary even between clouds of similar optical thickness. This makes it even more difficult to distinguish optically thin clouds from a glittery ocean surface on a per-pixel basis and thus is part of the cause for threshold based retrieval methods to miss those thin clouds. Previously, we set a "conservative" threshold of 2 to account for the uncertainty due to 3D biases. While it is reasonable to assume that almost no cloud above a cloud optical thickness (OT) of 2 would go undetected by a passive imager, in this study we are more interested in a threshold below which it is **likely not to detect** clouds using passive imaging. Changing the threshold from 2 to 1 for the current analysis changes the portion of optically thin clouds and thicker clouds in the WALES lidar dataset. In particulars the contribution of optically thin clouds to the total cloud cover decreases from 62% (< 2 OT) to 42% (< 1 OT).

**R1\_C1**

Question: Cirrus: The ASTER thermal band is not used, which would help to identify cirrus (on the expense of a lower resolution). Has it been tested, if there might be cirrus (especially optically very thin cirrus)? Because, the authors use radiative transfer simulations to estimate the cloud free state. Afterwards, they use ASTER pixels, which were clearly identified by the common ASTER cloud mask as cloud free, to adapt the aerosol optical depth (AOD) in the simulations. If there is optically thin cirrus, which is not considered or undetected by the cloud mask, I guess it would bias the adaption of the AOD. I suggest to elaborate a bit further, if there could be cirrus or not. In the case of yes it would be good to know how much such a cirrus might bias the method of identifying optically thin clouds. Even, if there was less or no cirrus during EUREC4A, the discussion about cirrus would help to decide, if this method can be easily transferred to regions somewhere else on the globe.

Answer: Yes, cirrus can impact the retrieval just like the reviewer pointed out. We checked two ways of detecting cirrus in the current dataset. First, we used a methodology to detect cirrus/thin clouds within ASTER's thermal bands following the publication by [Hulley and Hook, 2008]. And second, we looked at the co-located MODIS cirrus reflectance flag product.

Both methods detect selected obvious cirrus cases from the dataset (manual visual check). However, the MODIS cirrus reflectance flag product has a 1km pixel resolution and shows to be less suited in scattered trade cumulus cloud fields as the small clouds mostly result in "flag 2:cirrus pixel" while the high-resolution ASTER cloud mask clearly shows small trade cumulus clouds in those regions. We therefore focus on the Hulley and Hook method. The authors describe their method as follows:

"the thin-cloud/cirrus test utilizes a brightness temperature difference test between ASTER bands-13 (10.6 mm) and 14 (11.3 mm) (BT10.6 - BT11.3) and works well for thin cirrus, the edges of thicker cloud, low reflectance clouds and a variety of other types of cloud. This test was based on the AVHRR approach developed by Saunders and Kriebel [1988]."

As described and also noticed when applying this algorithm to the ASTER EUREC4A dataset, the "cirrus/thin cloud" flag detects the few visually selected actual cirrus-cases, but mostly it marks areas close to cloud edges. Generally, we can assume that cirrus clouds are extended, larger cloud structures. If present, we would expect a significant fraction of the non-cloudy (ASTER cloud mask) areas to be flagged by the Hulley and Hook method. Iteratively changing the threshold in potentially cirrus-contaminated area outside of detected cloudy areas, the relevant results from the current study change as follows:

| Allowed
potential
cirrus
amount /
% | Images
remaining
from 395 | OTC
cover | Detected
cloud
cover | OTC
contribution
to total
cloud
cover / % | OTC
reflectance | Detected
cloud
reflectance | CRE
bias
/ % |
|-------------------------------------------------|---------------------------------|--------------|----------------------------|-------------------------------------------------------|--------------------|----------------------------------|--------------------|
| 100                                             | 395                             | 14.5         | 28.7                       | 33.5                                                  | 0.058              | 0.208                            | -7.7               |
| 30                                              | 390                             | 14.2         | 28.7                       | 33.1                                                  | 0.058              | 0.208                            | -7.6               |
| 10                                              | 380                             | 14.1         | 28.5                       | 33.1                                                  | 0.058              | 0.209                            | -7.5               |
| 5                                               | 365                             | 14.0         | 27.7                       | 33.6                                                  | 0.057              | 0.210                            | -7.4               |
| 1                                               | 309                             | 13.8         | 23.9                       | 36.6                                                  | 0.056              | 0.209                            | -7.5               |

Most importantly, we notice that the numbers change only marginally indicating that cirrus does not have a strong impact on the current study. Nevertheless, we can exclude some of the cirrus cases by working with the threshold of 10% which leaves 380 images for the current study.

We added a respective paragraph in section 2.1 where the ASTER dataset is described.

**Minor comments:**

**R1\_C2**

**Acronyms**: Acronyms are often used several times before they are introduced the first time. Examples are EUREC4A, WALES, HALO. I don't know if I got them all. Please check all acronyms throughout the manuscript and introduce their full names whenever they are used for the first time.

Answer: Done. Thanks for making us aware of it!

**R1\_C3**

**Clear-Sky**: It should be rather called cloud free. Clear-sky would also mean aerosol free. I would suggest to check it throughout the whole manuscript and exchange clear- sky by cloud free wherever it is appropriate.

Answer: We agree, "cloud-free" is clearer :) we leave the flag CLEAR when talking about the Simplified Clear-Sky Model (SCSM), but change the usage of clear-sky in the text to cloud-free.

**R1\_C4**

**Indices and units:** Indices are sometimes written in italic letters and sometimes in non-italic letters. Throughout the manuscript this happens also for one and the same index (as an

example Eq. A1 and Eq. A2). For reasons of consistency you should write all indices in nonitalic letters.

Answer: We changed all subscripts to non-italic letters.

**R1\_C5**

**Fig. 1 and L70-L71**: This figure is not showing the measurements itself, but rather the position, where they were taken.

Answer: Yes, we reworded the text to say "Fig. 1 shows the location of measurements taken in the area east of Barbados[...]" and changed the Figure caption accordingly.

**R1\_C6**

**Fig. 4**: The dark blue line is really hard to identify. Maybe it helps to draw it in red color to increase its contrast compared to the black line. Furthermore, compared to the inset figure, the number of only 8 % seems to be too small. But this might be only an optical illusion due to the distribution of the clouds.

Answer: We changed the color for ASTER to a brighter one with a better contrast to black. Due to the described changes in General part in the beginning of this document the ASTER cloud mask cover was indeed slightly underestimated. Based on the corrected dataset, the value is 0.1 instead of 0.08. Nevertheless, the small plot with white cloudy pixels surrounded by black pixels visually enhances the "cloudy" area. The inset figure covers the same reflectance range as shown in the reflectance distribution, i.e. from 0 to 0.15. From the reviewers comment we assume that reducing the reflectance range can be misleading and we changed the colorbar such that it shows reflectances from 0 to 1. See OLD (top) and NEW (bottom) versions on the right.

Figure: OLD version submitted previously on top and the corrected version on the bottom.

**R1\_C7**

**P7, L154-L155**: ... select 20000 pixel ...; Just from the number it is hard to estimate, if 20000 pixel are a lot or not. Maybe it is worth to include some information on the total pixel number per image in Sect. 2.1. I might be wrong, but I think it should be 5000 pixel per swath, but only three quarter of them are used depending on the viewing direction.

Answer: 20000 pixels are ~0.11% of valid image pixels (excluding swath edges). We added this information in section 3.1 and additionally elaborated on the total valid amount of ASTER image pixels in section 2.1 by adding "One image of band 3 radiances consists of 4200 pixels along track and 4980 pixels across track where, depending on the viewing angle, about 15.4\,\% are swath edge pixels and neglected within the further analysis leaving about 17684552 pixels per image."

**R1\_C8**

**Tab. 1 and Tab. 2**: I would suggest to combine both tables. The header is already the same. Otherwise it is confusing, why there are two tables, with identical header, but different values. If the authors prefer to keep two tables I would suggest to include at least the parameter description (Dp(OTC), DE(R|OTC)) also in table and not only in the table caption.

Answer: That's a good point and we merged the tables as suggested.

**R1\_C9**

**Tab. 3**: The authors should include a third row indicating the cloud cover, which was originally derived by the ASTER cloud mask and probably name the first row "undetected optically thin cloud cover". Of course, it can be easily calculated from the difference, but it avoids confusion (different number in L319, if I calculate it from the table) and helps to highlight the large number of the so far undetected optically thin clouds.

Answer: We added one further decimal place to the numbers to avoid misunderstandings through rounded numbers. Also, we added the suggested additional column stating the numbers for the "detected, thicker" clouds in the ASTER cloud mask and for WALES for clouds with cloud OT > 1. The new table looks as follows:

**Table 2.** Cloud cover estimates during EUREC4A from 395 ASTER satellite observations  $(60 \times 60 \text{ km}^2)$  at 15 m resolution on 17 days andfrom WALES lidar measurements recoded within 13 research flights (days) at about 40 m resolution in January and February 2020.

|                | Undetected optically thin cloud cover / % | Detected (*)
cloud cover / % | Total cloud cover / % |
|----------------|-------------------------------------------|--------------------------------------------|-----------------------|
| ASTER (mean)   | 14.1                                      | 28.5                                       | 42.6                  |
| ASTER (median) | 13.3                                      | 16.7                                       | 34.9                  |
| WALES (mean)   | 14.3                                      | 19.3                                       | 33.7                  |

 $^{(*)}$  "detected" refers to the ASTER cloud mask and in the case of WALES data to clouds with cloud optical thickness  $\geq$  1.

**R1\_C10**

**Fig. 9:** The section (L361-L372) related to Fig. 9 and Fig. 9 itself needs to be revised. Please include (a) and (b) in the panels. Maybe it is a good idea to include headlines in the single panels. Otherwise it is hard to understand what the single panels are about and why the two panels are different. Also from the related section it became not clear to me.

Answer: We added information to Fig. 9 as well as to the paragraph pointed out by the reviewer. We agree, that this is hard to understand in general. Nevertheless, it

shows so nicely the differences in what we define to be a "cloud" and how cloud brightness changes with the respectively defined cloud cover, that we would like to keep it in and take the risk to loose some of the readers for a second.

**Technical comments:**

All technical comments are taken care of and mostly implemented as suggested.

Question: The ASTER cloud mask identifies pixels in four categories: confidently clear, probably cloudy, and confidently cloudy pixels. In this study, a simulated clear- sky reflectance distribution constrained by ASTER cloud mask "confidently clear" pixels is used to identify optically thin clouds. By design, it seems that this approach assumes pixels identified as "probably clear" are optically thin clouds, which may not necessarily be true, especially in the vicinity of pixels that are probably or confidently cloudy. As a result, the authors may be overestimating the impact of what they are calling "optically thin clouds" since an unknown fraction of these may in fact be clear (e.g., humidified aerosols). There is no attempt to quantify the fraction of "optically thin clouds" that are clear (humidified aerosols). While this is likely not possible using the ASTER measurements, I believe the WALES lidar measurement can provide some relevant information, at least in the region and on the days WALES measurements are available.

Answer: The flag "probably clear" in the ASTER cloud mask and the flag "optically thin cloud (OTC)" defined within the current study are not the same. Speaking physically, we attribute OTC not to the normal mode of humidified aerosols always present in a humid boundary layer such as in the trades. As we pointed out in the first paragraph of section 5.2, humidified aerosols are part of the cloud-free signal in ASTER as well as WALES measurements. Instead, the OTC labeled areas in our analysis may include **anomalously** humidified aerosols within the continuum from clear to "detected" and typically more extensive clouds. However, we do see a possibility that fossil clouds, in the form of lingering pockets of humidified aerosol, might be classified as OTC. We think the WALES analysis, and the magnitude of the observed optical depths (from WALES) excludes this as a major contributor. Even if this inference was incorrect, we believe it would be more correct to think of cloud fossils as optically thin (and fading) clouds than as an aerosol signal, particularly since such signals will not scale with aerosol amount. We moved part of the information from section 5.2 to the introduction of the paper to make that clear from the beginning on and additionally added further thoughts in section 5.2.

In technical terms, we would like to clarify that the main difference between the ASTER and our approach lies within the definition of cloud-free areas. Through the use of a cloud-free RT model and analysis of the reflectance distribution function in stead of per-pixel analysis, we are able to look at reflectance differences of thin clouds within the noise of measurements (Fig.6, cloud 3D effects, sun glitter etc...) and thus improve on quantities representative to an observed scene. The tradeoff is that we can't directly attribute our cloud-free and OTC flags on a per-pixel basis. Thus, a direct comparison of OTC and "probably clear" pixels is only possible in a statistical sense, which is handled in **R2\_C3**.

**R2\_C2**

**Question:** The clear-sky methodology introduced in Section 3 and the ASTER cloud mask of Werner et al. (2016) are not independent since the ASTER cloud mask's "confidently clear"

population is used as input to the "clear-sky" method. It seems like the "probably clear" population from the ASTER cloud mask likely makes up the bulk of what is classified as "optically thin clouds" from the "clear-sky" method. If the "confidently clear" population from the ASTER cloud mask were too conservative or not conservative enough, how would this impact the results? That is, what is the sensitivity of the final results (fraction of undetected optically thin clouds) to the thresholds used in the ASTER cloud mask?

Answer: We use the cloud masking algorithm following Werner et al., 2016, as a reference for traditional thresholding tests. We do not aim to improve this algorithm within the current paper and because a sensitivity of our results to the cloud mask does not impact our conclusion (that OTC are important) we do not change any thresholds chosen therein. Certainly this should be explored if one aimed to improve on the Werner et al., 2016 methods.

**R2\_C3**

Question: What fraction of the pixels identified as "optically thin cloud" by the methodology in Section 3 are classified as confidently clear, probably clear, probably cloudy, and confidently cloudy pixels by the ASTER cloud mask? I suspect close to 100% fall into the "probably clear" category, but it would be interesting to know.

Answer: Within the current study, a pixel is not directly linked to a single flag value but rather has a certain probability for either of the flags. This probability is influenced by the appearance (i.e. reflectivity) of the pixels. Accordingly, we can not identify individual cloud-free or OTC pixels, instead we can only give a statistical answer. Assuming no artificial bias towards any category, the contribution to optically thin clouds is on average split into 54.6% from "confidently clear" and 45.4% "probably clear" pixels.

Figure 1: heatmap of the ASTER cloud mask flags (x-axis) and the flags of the current study, clear – optically thin cloud (otc) - cloud (y-axis).

**Question:** The WALES comparisons to ASTER would be more meaningful if the same days and region were considered in both cases. Table 3 compares 395 ASTER regions over 60x60 km^2 with WALES from 13 research flights (days) and over a limited spatial domain. Given the spatial and temporal variability of trade cumulus, this does not seem like an apples-to-apples comparison. Is there any reason why the authors can't make a more direct comparison (e.g., compare the same days and location)?

Answer: ASTER and WALES unfortunately don't have more closely colocated measurements during EUREC4A. The trades are a region of rich and highly variable cloud structures as nicely illustrated by recent publications such as Stevens et al., 2020 and Schulz et al., 2021. We would like to include all types of trade cumulus clouds and therefore evaluate the ASTER dataset statistically in a bulk approach to be able to make a statement as representative as possible for the trades. We did select measurements of ASTER and WALES whenever both instruments sampled in a similar area the same cloud pattern (manual check by eye from satellite images and platform locations as shown in the AERIS leaflet tool). All results provided within the paper do no change significantly and especially the main conclusions stated do not change with subsampling for colocated measurements. We therefore continued to work with the full dataset.

**R2\_C5**

Question: There is a danger that readers might be led to believe that all cloud masking algorithms share the same biases as the ASTER cloud masking algorithm of Werner et al. (2016). This may very well be the case, but it has not been shown in this paper. As a result, the paper needs to clearly state that the results shown here only apply to the ASTER cloud masking algorithm.

Answer: We agree that the aim of the paper is not testing cloud masking algorithms (see also answer to R2\_C2). Instead, we want to highlight the knowledge that can be gained when using the described approach compared to cloud masking algorithms that work with thresholding tests. The algorithm following Werner et al., 2016, is to our knowledge the most sophisticated, tested, and published cloud masking scheme for the ASTER data that works with thresholds. It is comparable to typical approaches used by Landsat or MODIS but with the limited possibilities that the few ASTER bands provide. As stated in the paper, we see the Werner-algorithm as a reference for typical thresholding tests, while we don't want to claim that there is no better way based on thresholding tests. We will rewrite relevant phrases in the paper to make that clear.

**Specific comments**

**R2\_C6**

Comment: Title of paper: Consider adding "low" or "shallow" before "clouds" in the title. Answer: We prefer the simpler title, as the point of the title is to alert the reader to the general point of the manuscript, and the scope is better articulate in the abstract.

Comment: Line 6: "In this study we develop a method to quantify the cloud cover from a clear-sky perspective." At this point in the paper (abstract), it is not at all obvious what this means. Consider revising or removing this sentence.

Answer: The sentence pointed out is followed by two sentences that describe the meaning in more detail. Of course, the details follow in the Methods section. We think it is reasonable to keep the sentence.

**R2\_C8**

Comment: Line 9: "common cloud masking algorithms". Only the ASTER cloud masking algorithm of Werner et al. (2016) is considered in this paper. "Common cloud masking algorithms" is therefore misleading and should be replaced with "the ASTER cloud masking algorithm".

Answer: We do see the ASTER cloud mask as a reference and representative for common thresholding cloud masking algorithms (see also our answers to **R2\_C5** above). We don't see the abstract to be a good place for discussing the details of the ASTER cloud masking approach, while we do agree with the reviewer and make explicit statements at other locations in the paper.

**R2\_C9**

Comment: Lines 9-10: "We find that the cloud-mask cloud cover underestimates the total cloud cover by a factor of 2." This should be replaced with: "We find that the ASTER cloud-mask cloud cover underestimates the total cloud cover by a factor of 2."

Answer: see R2\_C8.

**R2\_C10**

**Question:** Line 11: "a high abundance of optically thin clouds". Please specify whether these are optically thin low clouds or high clouds (or both).

Answer: We refer to optically thin **low** clouds only. Within the frame of this review process, we excluded possible images with cirrus (optically thin high clouds) from the analysis and changed the above sentence accordingly.

**R2\_C11**

Comment: Lines 17-18: Earth's trade wind regions combine a dry atmosphere and a high abundance of shallow clouds" What about thin cirrus above the shallow clouds? Is this not a common feature? How is thin cirrus screened from the ASTER analysis?

Answer: we refer the reviewer to our answer to **R1\_C1**

**R2\_C12**

Comment: Line 20: "Changes in the cloud radiative effect with warming pace cloud feedbacks" This phrase is unclear. Please reword.

Answer: we reworded this sentence to "Changes in the cloud radiative effect with warming can amplify or dampen global warming."

Comment: Lines 30-31: "Estimating the cloud cover is a well-known issue in the sense that it decisively depends on the instrument used and the purpose of respective datasets." The paper below could be cited here (in fact, I'm surprised that it was not cited at all): J. Stubenrauch et al., "Assessment of global cloud datasets from satellites: Project and database initiated by the GEWEX radiation panel," Bull. Amer. Meteorolog. Soc., vol. 94, no. 7, pp. 1031–1049, 2013. doi: 10.1175/BAMS-D-12-00117.1.

Answer: Indeed, the Stubenrauch paper fits very well here. Thanks for pointing out! We added a citation in line 42 in the revised document.

**R2\_C14**

Comment: Lines 44-47: The method used here ("clear-sky approach") depends upon the ASTER cloud mask clear-sky distribution as a starting point. As a result, It is not independent of existing cloud thresholding approaches.

Answer: This is correct. However, we extract the part of the ASTER "confidently clear" distribution that cannot be explained by cloud-free RT relations. That is the contribution of our approach which results in a significant amount of optically thin clouds being detected within the ASTER "confidently clear" distributions as shown in **R2\_C3**.

**R2\_C15**

**Comment:** Lines 52-53: "With the clear-sky approach we can detect enhanced reflectances from anomalously humidified aerosols and optically thin cloud areas that are undetected by traditional cloud-masking algorithms."

How do you distinguish between these two? Are humidified aerosols classified as optically thin clouds instead of clear? The humidified aerosol cases are likely classified as "probably clear" by the ASTER cloud mask and (presumably) optically thin cloud using the methodology described in Section 3.

**Answer: see R2\_C1**

**R2\_C16**

Comment: Line 53: "With the clear-sky approach we can detect enhanced reflectances from anomalously humidified aerosols and optically thin cloud areas that are undetected by traditional cloud-masking algorithms". Some groups use both approaches (e.g., Trepte et al., 2019).

Answer: We would like to thank the reviewer for pointing us to the paper. The CERES products are indeed the most sophisticated and operational products we know of. In our study, we do not claim to compete with such products, but rather want to learn something about trade cumulus clouds when looking at high resolution data.

**R2\_C17**

Comment: Lines 68-69: "a Sun-synchronous orbit with an equator crossing time of 10:30 local solar time"

"a **descending** sun-synchronous orbit with an equator crossing time of 10:30 local solar time"

Answer: we added "descending".

**R2\_C18**

Comment: Line 75: Cite Werner et al. (2016) reference after "ASTER cloud mask". Answer: we added the reference as suggested.

**R2\_C19**

Comment: Lines 76-77: "We further draw comparisons to the ASTER cloud mask which is based on several bands in the VNIR."

The two methods are not independent since the ASTER cloud mask's "confidently clear" population is used as input to the "clear-sky" method described in Section 3. It seems like the "probably clear" population from the ASTER cloud mask likely makes up the bulk of what is classified as "optically thin clouds" from the "clear-sky" method. The "optically thin clouds" are likely a mixture of humidified aerosols, optically thin low clouds and (possibly) optically thin high clouds.

If that is the case, the authors should point this out. As written, it sounds as if the two methods are independent.

Answer: The details and dependencies are stated in more detail in the methods section 3. Here, we want to introduce the data. We refer the reviewer to our answers to **R2\_C3** for the flags and **R2\_C1** for the aerosols.

**R2\_C20**

**Question:** Lines 89-92: Did WALES detect any thin cirrus above the trade cumulus in any of the scenes? This should be noted.

Answer: The HALO aircraft few at about 9km altitude most of the time. Cloud tops detected by WALES were all below 6km. Cirrus clouds are typically at higher altitudes and most likely above the aircraft if present. We added a respective sentence in section 2.2.

**R2\_C21**

Comment: Line 117: "lowest pressure level 1000 hPa". Should this be lowest ALTITUDE pressure level?

Answer: yes, we added "altitude".

**R2\_C22**

Question: Lines 133-135: "Knowing the theoretical clear-sky contribution to an all-sky ASTER image we can then investigate the cloud-related contributions that are undetected by the cloud mask and which we attribute to optically thin clouds." In broken cloud conditions, part of the observed clear-sky radiance is contaminated by scattering from cloud layers adjacent to cloud-free areas, which then redirect some of the scattered light into the sensor field-of-view. This results in a positive "clear-sky" radiance bias. This effect likely is not accounted for in the theoretical clear-sky calculation, so that some clear areas might be erroneously flagged as cloudy. Is there a sense of what the magnitude of this bias is? This should at least be mentioned here and maybe discussed in more detail later in the paper.

Answer: Indeed, 3D cloud radiative effects include the scattering of sun light at cloud edge and may potentially lead to brighter surroundings of clouds. In line 133-135 we introduce the methodology. We discuss the negligible influence of 3D radiative effects in our study in the results section together with the WALES results (line 328). The WALES lidar measurements are insensitive to such 3D effects compared to an imager like ASTER. In our analysis we find a similar total cloud cover in both datasets (ASTER, WALES) and similar contribution from optically thin clouds for WALES cloud OT < 1, that is OTC. As stated in the manuscript, we therefore assume that possible 3D effects don't unduly influence our results. We added a note in the methodology section within the paragraph cited by the reviewer and refer therein to a later point in the paper where we discuss the 3D effect. Together with WALES information.

**R2\_C23**

**Question:** Also, how does one distinguish between thin low clouds, thin high clouds and humidified aerosols? It seems these are all grouped together as "optically thin low clouds", which may not be true in reality.

Answer: thin high clouds (cirrus) are excluded within the revised analysis (see **R1\_C1**), humidified aerosols and thin low clouds are distinguished according to our answer to **R2\_C1** above.

**R2\_C24**

**Question:** Section 3.1: Has the simplified clear-sky model (SCSM) been compared with a more sophisticated model to assess the uncertainty in the clear-sky radiance calculations? From Appendix A, it appears not.

Answer: Actually it has. For single cases the SCSM has been compared to the full radiative transfer model LibRadtran. For the same sensor-sun geometry, same surface wind speed, and the effective AOD (that we get from the optimization approach that we apply to the SCSM and the ASTER cloud-free reflectance distribution), the resulting reflectance distributions agree remarkably well though the SCSM includes several further approximations for example for the diffuse radiation.

We also considered using the full LibRadtran model but this was computationally too intensive to be included in an optimization approach similar to the one that we use for the SCSM. Given the uncertainty in input parameters (e.g. AOD, etc.), an overall improvement due to a more accurate model would not be warranted.

**Question:** Lines 149-150: "Although the aerosol load does not vary much within a 60 x 60 km2 ASTER image" This is only really true for completely cloud-free conditions, free of dust and smoke plumes. In broken cloud regions, the AOD certainly vary appreciably due to humidification of the aerosols. Please clarify.

Answer: Yes, we agree, but our assertion is not one about AOD, rather about our ability to infer an effective AOD from the cloud-free signal. A statement has been added that this does not preclude a contribution to OTC from pockets of humidified aerosol or evaporating cloud drops associated with decaying clouds. in section 5.2. See also our answer to **R2\_C1**.

**R2\_C26**

Question: Line 151: "We assume that the pixels labeled confidently clear in the ASTER cloud mask are a good first guess for clear-sky and shall serve as a reference for finding a suitable effective AOD such that the simulated clear-sky values are in close agreement with the selected ASTER pixel values." So, the simulated clear-sky reflectance distribution is really constrained by the ASTER cloud mask's "confidently clear" pixels? By doing so, do the authors implicitly assume that pixels identified as "probably clear" are optically thin clouds? How do you know this for sure? How would the results change if the ASTER cloud mask were less conservative and labled some of the "probably clear" pixels as "confidently clear"?

Answer: See answers to **R2\_C2** and **R2\_C3** above. Note that we use the confidently clear regions only to constrain the **shape** of the reflectance probability distribution of clear sky (and thereby the assumptions about SCSM input parameters), but we use both, confidently clear and probably clear regions to estimate the amount of cloud-free areas. Thus, if probably clear regions look like cloud-free regions, we treat them as cloud-free. Further, the answer to R2\_C3 shows that "confidently clear" regions are still the largest contributors to OTC regions.

**R2\_C27**

Comment: Lines 158-159: "We further optimize this image AOD value iteratively by minimizing the summed squared difference between simulated and observed reflectances." AOD for each pixel or for the entire distribution of pixels?

Answer: The effective AOD value refers to a value representative for the distribution belonging to a single image. We added a sentence in this paragraph to make that clear.

**R2\_C28**

Comment: Line 162: "From comparing simulated clear-sky reflectance distributions to the observed ones for fully clear-sky ASTER observations". This sentence is unclear. What does "fully clear-sky ASTER observations" mean. Does this refer to "confidently clear" and "probably clear" pixels? Please clarify. Answer: Yes, this is indeed unclear. We rephrased the sentence to "From comparing simulated cloud-free reflectance distributions to selected observed ones for manually checked and seemingly cloud-free ASTER observations, we find two things."

**R2\_C29**

**Question:** Lines 171-172: "However, the ASTER dataset is confined to a narrow set of sensor-sun geometries and outside of possible sun glint observations." Is a glint angle threshold used to define areas influenced by glint? If so, please specify the value.

Answer: We don't expect sun glint to be a problem as the reflected sun angle of specular reflection of sunlight into the sensor is above 23°. The glint angle is calculated following Yang et al., 2015. In Mieslinger et al., 2019 we showed that showed that sun glint is negligible for ASTER observations with a sun glint angle above 25°.

Figure 2: Histogram of smallest pixel reflection angle per ASTER image from the EUREC4A dataset.

**R2\_C30**

**Question:** Lines 174-176: "The output from our SCSM model provides us with a distribution of clear- sky reflectances p(R|FCLEAR,B), which is the probability distribution of reflectance values R given that they originate from clear-sky area with the flag F = FCLEAR and additional background conditions B".

Strictly speaking, isn't this just the same as the distribution you would get if only "confidently clear" pixels from the ASTER cloud mask were considered?

Answer: No. p(R|FCLEAR,B) is the theoretical distribution for a perfectly cloud-free scene as calculated by the model and might differ in its shape from the distribution of "confidently clear" pixels from the ASTER cloud mask. In particular, the latter differs from the "true" distribution in its extend towards higher reflectances as can be statistically also seen in Figure 1 above. It is exactly this visible deviation from the theoretical cloud-free distribution that was a starting motivation for the current study.

**Question:** Lines 183-185: "The darkest observed pixels originate form clear-sky ocean observations. Small cloud fragments and humidified aerosols slightly enhance the reflectance, though they are often undetected by cloud masking scheme."

This sentence is likely only partly true. The population identified by the ASTER cloud mask as "probably clear" likely is a mix of clear pixels in humidified aerosol conditions, unresolved low cloud fragments and thin cirrus. This paper appears to assume that anything identified as "probably clear" is optically thin low cloud, without justification. As a result, it is likely that the bias attributed to the ASTER cloud mask is overestimated, but it is unclear by how much.

Answer: In the answer to R2\_C2 and the respective Figure 1 of this document we elaborate on the fact that the ASTER "probably clear" and the optically thin cloud flag are not the same. We would reword the sentence to say "decaying cloud fragments, which may also be manifest as lingering pockets of anomalously humidified aerosol" as the normal mode of humidified aerosols is part of the clear-sky signal (see also **R2\_C1**). Thin cirrus do not significantly impact our results, and as described in the answer to **R1\_C1** above, analysis has been performed to bound its effect.

**R2\_C32**

Question: Lines 203-204: "Thinking visually, we scale the simulated clear-sky distribution up until it touches the allsky distribution p(R)." What precisely does this mean?

Answer: A cloud-free fraction p(CLEAR)=0 would mean that the output of the SCSM is multiplied by 0. The cloud-free fraction is the factor that scales the distribution p(R|CLEAR) shown here on the right. An increasing cloud-free fraction scales the distribution p(R|CLEAR) up. However, the physical constraint that probabilities have to be positive limits p(CLEAR). This

limit is reached when with increasing p(CLEAR) the distribution p(R|CLEAR) touches the all-sky distribution p(R) at any point in the R space.The equation (6) and (7) are the mathematical basis, while the sentence marked by the reviewer is thought to help the people that like less equations but think more in plots and it might therefore not help everyone.

**R2\_C33**

Question: Figure 4: How does one interpret the light blue area to the left of the orange line? These are flagged as optically thin clouds whose reflectance is smaller than the clear-sky reflectance. How can this be? Also, it would be helpful if the authors used a different color for the "ASTER cloudy pixel" distribution. The two blue colors are hard to differentiate.

Answer: This is a plotting artifact as the probability of OTC in this range is 0 and the line has a defined width. We changed the color coding to enhance the contract between different lines.

**R2\_C34**

**Question:** Line 251: "and thus lead to strong over- or underestimation of p(OTC) as high as \_10 %." Please indicate if this is a relative bias or an absolute bias. I suspect it's the former.

Answer: correct, it's relative and we added that to the text.

**R2\_C35**

Question: Table 3: Why not compare the same days and the same location? Table 3 shows results of a comparison between 395 ASTER scenes over a large area and WALES for 13 research flights over a much smaller area (Fig. 1). It's not clear what conclusions can be drawn from this comparison since cloud variability is likely too large for these to be directly compared.

Answer: see answer to **R2\_C4** above.

**R2\_C36**

Question: Lines 330-338: Are the comparisons with the Leahy et al. (2012) study relevant at all. That study looked at two years for 60S-60N whereas the current study only considers the EUREC4A region for 17 days in January and February 2020.

Answer: we explicitly state the results by Leahy et al., 2012, for the trade wind region, where they find 84% coverage of optically thin low clouds. We set this in context to the WALES measurements from the EUREC4A period which are thought to sample typical trade wind conditions, too. Taking into account the uncertainty in CALIOP described in lines 330-338, the WALES and CALIOP measurements agree remarkably well.

**R3\_C1**

Question: A more detailed discussion on the sensitivity to some of the main controlling parameters. The authors do discuss sensitivities of the wind and the variance and show that they are expected to be low. However, since the residual estimation supposed to be super sensitive to the properties of the distribution, and especially to the variance, more information would be needed. For example, the authors use fixed variance, and intuitively I would expect the variance to be a function of the wind. Also, I miss the sensitivity discussion to the AOD estimations. I expect a tradeoff between errors in the ocean reflectance and the AOD that can explain the cloud-free reflectance distribution.

Answer: The current setup of the SCSM includes a dependency of the radiance on wind speed. And we include not an average wind speed, but a wind speed that is statistically "up-scales" by adding a variance in wind speed to the average ERA5 wind speed estimates that are available to each ASTER observation. We understand that the reviewer suggests that the additional variance in the radiance that we add on top of the (average) Cox and Munk parameterized radiance value on a pixel level would have an additional dependency on the wind speed. However based on the sensitivity studies we have performed we see no reason why this would materially influence our results.

**R3\_C2**

Question: On the same note - L. 162-170: Can variability in the AOD contribute to the clearsky distribution variance? Since the variance is estimated from clear-sky images (97% confidently clear), it can be biased to clear sky aerosol distributions. As the authors mention, aerosols and meteorological conditions (especially surface wind) tend to correlate. I suspect that the variability in AOD will be larger in convective days that permit clouds (due to meteorology and not due to cloud "twilight" effects). Can the authors include a sensitivity test regarding this issue or disavow the argument?

Answer: We agree with this interpretation. The reviewer's suggestion opens up a new line of inquiry which we find interesting, but not related to our central points and hence better developed in its own study.

Indeed, we would expect AOD to increase with surface winds as more sea salt particles may linger within the lower boundary layer. Meteorological conditions are thought to act on the **large scale** and will influence individual ASTER observations, but influence less the variability within a single image. Also, we would like to clarify that we work with an effective AOD that is thought to be representative for the conditions within a single ASTER image. As such, an increase in the effective AOD will shift the reflectance distribution consistently to higher reflectance values but not change the width of the distribution.

**R3\_C3**

**Question:** line 61: Please mention if all the analyzed images were of trade wind cumulus and if there were any steps taken to filter out cold clouds and optically thin cirrus clouds.

Answer: within the current review process we make an attempt on excluding optically thin cirrus clouds. See the beginning of this document and the answer to **R1\_C1**.

**R3\_C4**

**Question:** line 228: What does it mean conceptionally, can you explain? Wouldn't the pixels with cloud shadows (which are not considered by the SCSM) will be classified as optically thin clouds? Can this explain OTC contribution in the lowest reflectance values (<0.05) in Fig. 3?

Answer: Yes, exactly. The OTC contribution indicated by the blue dashed line at about 0.048 is an example of what we assume to be due to cloud shadows. Howeve

---

## Author Response (AR2)

Dear Mr. Dunkerton, dear editorial teams

The final manuscript has to tiny changes:
1. We included a note on the "editorial team" in the acknowledgements (new document line 655)
2. We updated the published dataset to the paper on zenodo and changed the respective DOI in the manuscript (new document line 530)

Within the ACP online tool we have to certify that the version before and the uploaded one are identical. Except for the two changes above everything is exactly the same and we therefore hope that it is ok to still call it "identical" as there is no other option available.

Thank you very much for your editorial work and we wish you all the best,
Theresa